

# Spring and summertime aerosol optical depth variability over Arctic cryosphere from space-borne observations and model simulation

Basudev Swain[1], Marco Vountas[1], Adrien Deroubaix[1,3], Luca Lelli[1,2], Aishwarya Singh[5],
Yanick Ziegler[1,4], Sachin S. Gunthe[5], and John P. Burrows[1]

[1]Institute of Environmental Physics, University of Bremen, Germany
[2]Remote Sensing Technology Institute, German Aerospace Centre (DLR), Wessling, Germany
[3]Max-Planck-Institut für Meteorologie, Hamburg, Germany
[4]Karlsruhe Institute of Technology, Institute of Meteorology and Climate Research-Atmospheric Environmental Research
(KIT/IMK-IFU), Garmisch-Partenkirchen, Germany
[5]Indian Institute of Technology Madras, Chennai, India

**Correspondence:** Basudev Swain (basudev@iup.physik.uni-bremen.de)

**Abstract.**

The Arctic is a unique part of the Earth system that is currently undergoing a warming phase called the Arctic Amplification (AA). Changes in aerosol abundance and composition are under the influence of the AA and may be, in turn, important drivers to the AA. However, their ground and space observations are particularly difficult and spatio-temporally sparse in this region, limiting the knowledge and ability to model their variability. In this study, we have used the total aerosol optical depth (AOD) determined by the AEROSNOW algorithm using data from the AATSR satellite instrument over snow- and ice-covered regions of the Arctic. This data is then used to evaluate the global GEOS-Chem 3D chemical transport model for the period 2003-2011. Thus, the main drivers of monthly and seasonal variations in spaceborne AOD were determined by using the GEOS-Chem model-simulated aerosol components. By comparing these two AOD datasets, we examined the spring and summer AOD over Arctic snow and ice for the period of space-borne observations. The space-borne and modelled AOD show consistent spatio-temporal distributions in both seasons, with a pronounced chemical speciation in GEOS-Chem. This behaviour is attributed to the different seasonal sources of AOD. In spring, Arctic aerosols originate from long-range pollution transport from low and mid-latitudes as well as from local sources, whereas in summer natural local sources within the Arctic Circle (here defined as $> 60°N$) dominate. Arctic AOD is generally highest in spring and lowest in summer due to wet scavenging. In addition, carbonaceous aerosols (black carbon, BC, and organic carbon, OC) are an increasingly important contributor to total AOD over Arctic sea ice in summer due to the expected increase in boreal forest fires. The relative contribution of sulfate to total AOD over Arctic sea ice decreases while that of carbonaceous aerosols increases during the spring-summer transition. This suggests that boreal wildfires are penetrating more deeply into Arctic sea ice at higher latitudes during this study period. GEOS-Chem showed a systematically smaller AOD value compared to AEROSNOW over the Arctic sea ice region in summer. The promising results of AEROSNOW could also serve as the baseline for the evaluation and improvement of aerosol forecasts for various chemical transport models, especially over Arctic sea ice.



## 1 Introduction

Over the past three decades, near-surface air temperature in the Arctic has increased by two to four times the global average (Budyko, 1969; Serreze and Francis, 2006). This phenomenon is known as Arctic Amplification (AA). This warming of the
Arctic has amplified the retreat of glaciers, sea ice, and snow-covered areas (Shukla et al., 2019; Dai et al., 2019). AA is influenced by several processes, e.g. surface albedo feedback (Perovich and Polashenski, 2012), warm air intrusion and oceanic heat transport (Boisvert et al., 2016; Nummelin et al., 2017), cloud feedback (Kapsch et al., 2013; He et al., 2019; Middlemas et al., 2020), lapse rate feedback (Pithan and Mauritsen, 2014), biological and oceanic particle emission effects (Park et al., 2015; Campen et al., 2022), and remote feedback processes (Huang et al., 2021).

Atmospheric aerosols are suspended solid or liquid particles that are emitted from both natural and anthropogenic sources. Neither the contribution of aerosols to AA nor the effects of AA on aerosol load and its components are well understood. In the Arctic, aerosols exhibit considerable variability in their physical properties, chemical composition, size, and concentration (Willis et al., 2018).

In this study, aerosol optical depth (AOD) was used as a parameter to measure aerosol abundance and assess its impact on
the Arctic climate and vice versa. The sources of aerosol in the Arctic include long-range transport and changing regional and local emissions of aerosol or its precursors (Willis et al., 2018). Changes in the scattering and absorption of incoming solar radiation by aerosol directly affect climate (Bond et al., 2013). Increased amounts of aerosols that scatter solar radiation back into space cool the atmosphere and surface. This effect is called the aerosol dimming effect (Ramanathan, 2007). On the other hand, the aerosol that absorbs solar radiation warms the atmosphere and surface (Willis et al., 2018). Aerosols also act as
both cloud condensation nuclei (CCN) and ice nuclei (IN), affecting the microphysical and radiative properties of clouds. In this way, aerosols also indirectly affect climate change (Twomey, 1977; Kaufman and Fraser, 1997; Hartmann et al., 2020). Carbonaceous aerosols absorb solar radiation when deposited on snow and ice surfaces, reducing the albedo of snow and ice (Willis et al., 2018). This increases the melting rate of snow or ice, at least until it is covered with fresh, clean snow (Zhang et al., 2004; Dang and Liao, 2019; Im et al., 2021).

During summer (June, July, August, (JJA)), the major source of carbonaceous aerosols in the high Arctic is the result of biomass burning within the Arctic (Willis et al., 2018). In contrast, during spring (March, April, and May, (MAM)), long-range transport of emissions, arising from anthropogenic pollution and including biomass burning at mid-latitudes, dominate the sources of aerosol in the Arctic (Willis et al., 2018). Natural aerosol sources are predominant in the Arctic summer. These sources include wind-blown dust over land, sea salt over the Arctic Ocean, and biomass burning(Willis et al., 2018).

The Arctic is vast, and the lack of spatio-temporally representative ground-based measurements of AOD limits our understanding of the aerosols on the Arctic climate and on AA in particular, and vice versa. Recently, several research campaigns/expeditions have taken place. Amongst other objectives selected processes relevant to aerosol formation and loss in the Arctic. Examples are the MOSAiC campaign (https://mosaic-expedition.org), ACLOUD/PASCAL (Wendisch et al., 2019), PA-MARCMIP (Hoffmann et al., 2012; Nakoudi et al., 2018; Ohata et al., 2021). In addition, there are other site-based long-term



aerosol measurement studies (Herber et al., 2002; Tomasi et al., 2007; Moschos et al., 2022; Schmale et al., 2022). However, these are not necessarily spatio-temporally representative of the high-latitude Arctic region (Xian et al., 2022).

These sparse AOD measurements are used in part to explain AOD variations in climate models (Sand et al., 2017; Palazzi et al., 2019). In addition, the lack of AOD measurements in the Arctic, which results in a gap in observational data, further limits our knowledge of aerosol-AA interactions in global and regional climate models(Goosse et al., 2018)

Given the lack of ground-based measurements, several attempts have been made to use AOD, retrieved from the top-of-atmosphere reflectance (TOA) observations made by passive satellite remote-sensing instruments over the Arctic, to fill this gap Glantz et al. (2014); Wu et al. (2016); Sand et al. (2017); Xian et al. (2022)). However, these studies were conducted over the open ocean and snow- and ice-free surfaces. These AOD products are of limited quality and coverage over the cryosphere due to insufficient surface reflectance parameterization (Mei et al., 2020a) and Arctic cloud cover (Jafariserajehlou et al., 2019).

In addition, the global AOD retrieval data are more difficult, because of the large illumination angles. This potentially leads to an overestimation of AOD values (Mei et al., 2013).

Several pioneering research studies have been undertaken to develop algorithms for retrieval of AOD over snow and ice using passive satellite remote sensing, such as, Istomina et al. (2011) and later Mei et al. (2013, 2020b, a). However, these attempts have been largely confined to the island of Spitsbergen in the Svalbard archipelago in northern Norway. Therefore, there have

been no attempts to systematically apply these algorithms to the entire Arctic cryosphere and thereby eliminate the data gap, described above. In addition, the successful active sensor, cloud-aerosol lidar with orthogonal polarization (CALIOP/CALIPSO), does not report measurements above 72° N latitude. This is a result of the minimum detection limit, which depends on the signal-to-noise ratio over the Arctic (Pitts et al., 2013; Manney et al., 2015; Toth et al., 2018; Xian et al., 2022).

Furthermore, in addition to ground-based measurements and satellite observations, several valuable studies using models

and reanalysis datasets have been conducted over the Arctic: examples being Sand et al. (2017); Breider et al. (2017); Stone et al. (2014); Ren et al. (2020); Xian et al. (2022). It is worth noting that all of these studies lack observational data, especially over the high Arctic cryosphere region, which is highly vulnerable to warming and climate change in the Arctic.

The scientific objective of this study is to use satellite observations of total AOD and analyze the AOD components, transport, meteorological conditions, and natural and anthropogenic aerosol sources that determine total AOD over the high Arctic

cryospheric regions. At the same time, model results were compared with AEROSNOW results in the absence of in situ and ground-based observations of AOD over high Arctic sea ice. AEROSNOW is the algorithm used to retrieve AOD over Arctic snow and ice by using the Advanced Along Track Satellite (AATSR), which is very well explained in Swain et al. (2023a). The former requires a good knowledge of the retrieval of the AOD distribution over the northernmost Arctic latitudes, which is well explained in (Swain et al., 2023a) and a previous study in Istomina et al. (2011). The latter requires a model that has an optimal

selection of AOD components. Such a component-based aerosol loading using passive satellite remote sensing measurements is currently available only for mid-latitudes and not for the Arctic.

As mentioned earlier, long-range transport of emissions from forest fires (Sand et al., 2017; McCarty et al., 2021) is an important source of carbonaceous aerosol, i.e., BC and OC in the Arctic. Changes in these components will alter the AOD and lead to changes in the radiative forcing (Stone et al., 2014).



According to recent studies (Sherstyukov and Sherstyukov, 2014; Hugelius et al., 2020; McCarty et al., 2021), biomass burning in the low Arctic will increase in the future. In this regard, peat thaw in Siberia is also of potential importance. Efforts to better understand the total AOD and corresponding aerosol components, especially in the vulnerable high Arctic, are therefore timely.

Consequently, we have extended our objective to examine aerosol composition over the snow- and ice-covered regions of the high Arctic (72-90°N, as defined by Sand et al. (2017)). To achieve this goal, we investigate aerosol composition in the high Arctic using GEOS-Chem, a global chemical transport model (CTM) simulation.

In this study, we determine the total AOD using an approach described in Swain et al. (2023a). In addition, we used the Goddard Earth Observing System 3-D global chemical transport model (GEOS-Chem) (GC) simulations (Bey et al. (2001)) to study and analyze the various aerosol components. These analyses use 3-D simulations adapted to Arctic conditions. For example, global and regional biomass-burning emission inventories (and more) are used in these models. The latter was updated and coupled with assimilated Modern-Era Retrospective Analysis for Research and Applications, Version 2 (MERRA2) meteorology.

We have generated the AEROSNOW AOD data set for the period from 2003 to 2011 (9 years) (Swain et al., 2023a). The algorithm uses the measurements, made by the Advanced Along-Track Scanning Radiometer (AATSR) data over the Arctic. To determine the quality of the AEROSNOW and the GC data sets, we compare with accurate ground-based Aerosol Robotic Network, AERONET, measurements in the high Arctic. The description of the AERONET dataset is given in section 3. The AERONET, AEROSNOW, and GC AOD are then compared at the high-latitude AERONET sites. and GC and AEROSNOW AOD over the high latitude snow and ice-covered surfaces (see section 3). In section 4 we show AOD retrieved from AATSR measurements and that simulated by GC models. The latter includes the component AOD. Finally, we draw conclusions in section 5.

## 2 Data Sets and Data Processing

To investigate the distribution, variability, and remote and local sources of Arctic aerosols over snow and ice, we have used passive remote sensing in spring (March-April-May, MAM) and summer (June-July-August, JJA). Comparison of AOD in these two seasons enables the impact of the long-range transport and local aerosol sources to be identified and investigated (Willis et al., 2018).

The AEROSNOW algorithm is applied to the dual view Level 1B reflectance at the top of the atmosphere made by the Advanced Along-Track Scanning Radiometer (AATSR) to retrieve AOD. More detailed information is given in (Swain et al., 2023a). We have validated and evaluated the retrieved AEROSNOW AOD and GC simulated AOD by comparison with the AOD, measured by the ground-based sunphotometer measurements, AERONET (Holben et al., 1998). Aerosol properties, components, and sources are estimated using the GC simulations. The study period is limited to 2003-2011, because this is the period for which AROSNOW AOD is available.



## 2.1 AERONET Level 2 Aerosol Product

The AERONET network of ground-based global sunphotometers measures solar and sky irradiance at various wavelengths from the near ultraviolet to the near-infrared with high accuracy (Holben et al., 2001; Giles et al., 2019). AERONET sunpho-

tometers record AOD values every 15 minutes in typically seven spectral channels (nominally 340, 380, 440, 500, 670, 870, and 1020 nm) (Holben et al., 2001). The quality-assured AERONET version 3 level 2 data are used in this study (accessed at http://aeronet.gsfc.nasa.gov).

The AOD from AERONET stations in the high Arctic was used to assess the data quality of AOD simulated by the GC model. The locations of the AERONET stations selected are OPAL (79.990°N, 85.939°W), Hornsund (77.001°N, 15.540°E),

Thule (76.516°N, 68.769°W) and shown in Fig. 1. Two sites are located over the Canadian archipelago (CA), which typically has aerosol of natural origin (Breider et al., 2017) and one station Hornsund on Spitsbergen, which is known to be affected by polluted air masses transported from lower latitudes.

Fine mode (FM) and coarse mode (CM) AODs at 500 nm using the Spectral Deconvolution Method (SDA) are used in the analysis of the AEROSNOW and GC AOD (O'Neill et al., 2003; Saha et al., 2010). To compare the FM and CM results

of AERONET measured at 500 nm with those of AEROSNOW (measured at 550 nm), wavelength conversion is required. However, tests have shown that the CM AOD at 500 nm may be assumed to be the same as that 550 nm value, while the FM spectral derivative at 500 nm is used to extrapolate the FM AOD at 550 nm. To compare the AOD of AERONET measured at 500 nm with those of GC (measured at 550 nm), wavelength conversion is similarly necessary. The derivative at 500 nm is used to extrapolate the AOD to 550 nm by using the 500-870 nm Angstrom Exponent. AERONET observations were then monthly

averaged and compared to values measured within a 25-km radius of the AERONET stations for the GC AOD. Monthly averages were calculated using the matched GC and AERONET AOD.

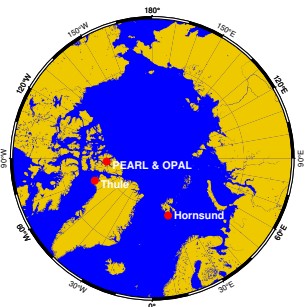

**Figure 1.** Location of PEARL(80.054°N, 86.417°W), OPAL(79.990°N, 85.939°W), Hornsund(77.001°N, 15.540°E) and Thule(76.516°N, 68.769°W) AERONET measurement stations considered in this study.



## 2.2 GEOS-Chem Model description

In this study, we use version 12.2.1 of the GEOS-Chem global 3-D model (http://acmg.seas.harvard.edu/geos/, Bey et al. (2001)) driven by 6-hourly assimilated meteorological fields from the National Aeronautics and Space Administration's (NASA)

Goddard Modeling and Assimilation Office's (GMAO) Modern Era Retrospective Reanalysis2 (MERRA2). The fully coupled model simulations included O3-NOx hydrocarbon chemistry, aerosols, and gas-aerosol phase partitioning (Alexander et al., 2005; Hu et al., 2007; Fountoukis and Nenes, 2007; Knippertz et al., 2015). As described in (Breider et al., 2017), aerosol simulations in GC include the following aerosol components: black carbon (BC), organic carbon (OC), sulfate-nitrate-ammonium, dust, and sea salt. The simulation of carbonaceous aerosols such as BC and primary OC (POC) are made using standard GC

simulations described by Park et al. (2003). The model assumes that 80% of BC and 50% of POC emitted are hydrophobic and the remainder is hydrophilic. After an e-folding aging time of 1.15 days the hydrophobic BC and POC are converted to hydrophilic components by Park et al. (2005), which can then be removed by wet deposition. The formation of sulfate aerosol occurs by the oxidation of $SO_2$. Initially OH reacts with $SO_2$ to make $HSO_3$, which reacts with $O_2$ to make an $HO_2$ and $SO_3$. The latter reacts in the gas phase with $H_2O$ to form $H_2SO_4$.

In GC model, we have used different processing schemes, such as the aerosol wet deposition scheme (Liu et al., 2001), the dry deposition scheme (Fisher et al., 2011), the dust mobilization scheme for wind speed subgrid variability (Ridley et al., 2013), sea salt aerosol simulation (Jaeglé et al., 2011) and optical aerosol properties (Koepke et al., 1997; Drury et al., 2010), and a linearized climatological ozone parameterization for stratospheric ozone (McLinden et al., 2000).

The simulation used a 10-min time step for transport and a 20-min time step for chemistry and emissions, with a horizontal

resolution of 4° × 5° (about 440 km × 100 km at the high Arctic latitudes of OPAL) and 72 vertical levels up to 0.01 hPa (Bey et al., 2001; Lu et al., 2020), for the period from 1999 to 2011. 1999 to 2001 is the model spin-up period.

Using the different components of AOD given by GC, the simulated AOD are divided into fine and coarse mode components ($\tau_{f,GC}$ and $\tau_{c,GC}$). The components comprise fine-molecular organic carbon (OC), sulfate ($SO_4$), and BC with fine- and coarse-molecular sea salt (SS) and fine- and coarse-molecular mineral dust (Hesaraki et al., 2017). The coarse and fine mode AOD are

then given by:

$$\tau_f = \sum_{l=1}^{72} (\tau_{f,l,SO4} + \tau_{f,l,BC} + \tau_{f,l,OC} + \tau_{f,l,SS} + \tau_{f,l,dust}), \qquad \tau_c = \sum_{l=1}^{72} (\tau_{c,l,SS} + \tau_{c,l,dust}), \qquad (1)$$

with l being the 72 vertical levels.

The total AOD was calculated by GC at 550 nm using optical properties from the global aerosol data set (GADS) (Koepke et al., 1997) with updates from more recent observations (Drury et al., 2010). The GADS comprises wavelength-resolved

complex refractive indices, and estimates of aerosol size distributions in terms of geometric mean and standard deviation at eight different values of relative humidity (0, 50, 70, 80, 90, 95, and 99%) (Martin et al., 2003). This information is used as input to the Mie software (Mishchenko et al., 1999), which then generates the optical properties assuming a lognormal distribution. The extinction efficiency ($Q_{ext}$) and effective radius ($r_{eff}$) are estimated. These are then used for the AOD calculations described



in (Martin et al., 2003). The AOD was calculated using the following equation:

$$\tau = \frac{3}{4} \frac{Q_{ext} M}{r_{eff} \rho}$$


(2)

Where the column mass loading and the particle mass density are presented as M and $\rho$ respectively (Tegen and Lacis, 1996).

### 2.2.1 Emission inventories used

The GC emissions were configured using the Harvard-NASA Emissions Component module (Keller et al., 2014). Global anthropogenic emissions of species include aerosol (BC, OC), aerosol precursor and reactive compounds ($SO_2$, NOx, $NH_3$,

$CH_4$, CO, NMVOC) as well as $CO_2$ and were taken from the Community Emissions Data System (CEDS) inventory (Hoesly et al., 2018). Monthly mean aircraft emissions were taken from the Aviation Emissions Inventory v2.0 (AEIC) (Simone et al., 2012), the inventory for biofuel and agricultural field burning in the developing world was taken from (Yevich and Logan, 2003). The US-American and Mexican inventory (BRAVO, Mexico Bend Regional Aerosol and Visibility Observational study) was used (Kuhns et al., 2005).

The latest anthropogenic ammonia ($NH_3$) emissions inventory over Canada was provided by Agriculture Canada, acquired from detailed surveys on monthly emissions of five agricultural categories: beef, dairy, fertilizer and poultry from APEI inventory (Sheppard et al., 2010). In addition, the Co-operative Programme for Monitoring and Evaluation of the Long-range Transmission of Air Pollutants in Europe (EMEP) anthropogenic emissions inventory for Europe (EMEP) was used (Auvray et al., 2007). Natural, biofuel, bird colony, and oceanic $NH_3$ emissions were taken from the Global Emission Initiative (GEIA)

inventory (Bouwman et al., 1997; Croft et al., 2016). Additionally, the National Emissions Inventory produced by the US EPA (EPA/NEI2011) (Simon et al., 2010), anthropogenic VOC emissions from (RETRO) (Bolshcer et al., 2007), MIX Asian emission inventory for emissions over south Asia (Li et al., 2017), DICE-Africa anthropogenic emissions inventory (Marais and Wiedinmyer, 2016). Non-anthropogenic emissions include biomass burning emissions. They were taken from the Global Fire Emissions Database version 4 (Giglio et al., 2013), volcanic sulfur dioxide (SO2) emissions (Fisher et al., 2011), Sea Salt (SS)

aerosol (Jaeglé et al., 2011), and mineral dust (Zender et al., 2003; Fairlie et al., 2007). Hence, dust and sea-salt fluxes in the model are independent of the anthropogenic, biogenic, and pyrogenic emission inventories used for other species. Emissions from other natural sources (e.g., lightning sea flux, soil-NOX) were also included (Fisher et al., 2011).

During the study period (2003-2011), the total AOD over the Arctic may have been influenced by stratospheric volcanic contributions from the Kasatochi and Sarychev eruptions in August 2008 and July 2009, respectively. Smaller eruptions into

the troposphere may also contribute to the total AOD. However tropospheric ash and sulfate aerosol, produced by volcanic eruptions, is typically much shorter-lived than the aerosol, produced by volcanic eruptions into the stratosphere. GC takes account of these eruptions by using the (Fisher et al., 2011; Carn et al., 2015) inventory.

According to (Sawamura et al., 2012), the contribution of stratospheric AOD is 0.01, i.e. up to about 25% of background but smaller than the AOD from anthropogenic sources such as the Arctic.

Increasing shipping traffic is likely to be a growing local contributor to the total AOD column (Gilgen et al., 2018; Raut et al., 2022). Shipping routes within the Arctic have increased during the Arctic amplification (Mudryk et al., 2021). The emissions



from shipping also act as a potential small source of carbonaceous aerosols to the Arctic (Browse et al., 2013). To represent this, ship emissions are taken from CEDS SHIP (Hoesly et al., 2018) and EMEP SHIP (Hoesly et al., 2018).

The main output of GC simulations in this study is the AOD values of the aerosol components: sulfate (SO4), black carbon
(BC), organic carbon (OC), sea salt in accumulation mode (SALA), sea salt in coarse mode (SALC) and dust. Further, the natural AOD has been simulated without any anthropogenic emission inventories.

## 3   Results: Evaluation of AODs from AERONET, AEROSNOW, and GEOS-Chem

Here we analyze the temporal evolution of the AEROSNOW results in comparison to station data. We also compare the collocated temporal results with GC model results. The time-series of retrieved AEROSNOW AOD is shown together with GC
modeling results and AERONET data in Fig. 2. GC is shown in a stacked fashion for each aerosol component. The topmost part of the stacked GC-AOD represents the total AOD modelled with GC. In general, the time-series for AERONET is well reproduced by both, AEROSNOW and GC. The best agreement occurs over PEARL station. Here AEROSNOW and GC agree with AERONET datasets well, with R values being 0.90 and 0.90, respectively. The latter is surprising bearing in mind, the cooperatively low spatial resolution of GC model results as compared to the point measurement made at AERONET stations.

In general and as expected, we can state that AOD at PEARL, OPAL and Thule (extended Canadian Archipelago, henceforth called CA-stations) exhibit similar temporal behaviors and also the partitioning of the AOD GC-components is similar at these stations. Hornsund is in this context clearly different and beyond a strong sulfate contribution the GC components show higher contributions of sea salt (SALA) and dust. The CA-stations show small average AOD in all three data sets. We attribute this to these stations having Arctic background conditions. One important difference is that in summer the CA stations exhibit on
average even higher levels of OC AOD than those at Hornsund. This is also reflected in plots of the seasonal averages over the four stations (Fig. 3). On average, all three datasets include some haze episodes during spring.

Having gained confidence in the satellite dataset (a detailed description is given in Part-I of this work), we now investigate the GC model results. In the analysis of Fig. 2, the overall monthly mean values from 2003 to 2011 over the four AERONET stations from GC simulations often show higher values than the ground-based observations. The most likely reason for this
is that GC simulates AOD independent of meteorological conditions, cloud cover, and the spatial and temporal constraints of the underlying bright surfaces, whereas the AEROSNOW retrieved AOD automatically impacted by these factors. Monthly averages are used for model evaluation with ground-based data because monthly averages significantly reduce model noise. The evaluation of GC AOD with AERONET and AEROSNOW and the respective Pearson correlation coefficient (R) and Root mean square error (rmse) values are shown in Fig. 2. The R value between the GC and AERONET AOD is higher than the
AEROSNOW because of variation in spatio-temporal availability of datasets Fig. A1, Fig. A2, Fig. A3, i.e. the continuous availability of GC AOD, whereas AEROSNOW depends on having cloud-free scenes.

In addition, AOD from an anthropogenic activity are large during spring, whereas naturally occurring AOD in the Arctic is dominant in summer. This result agrees with a previous study by Breider et al. (2017).





GC simulates AOD values, which agree well with AERONET and AEROSNOW AOD over all the four AERONET stations.
We conclude that accurate emission inventories and optical properties were used in GC, adequately representing the true AOD over these observation sites. The remaining biases of GC AOD are attributed to the coarse spatial resolution (4° × 5°). R-values can be improved by using finer nested grid simulations (0.5° × 0.666°), which would be suitable to capture the high values of short-lived aerosol loads (Yu et al., 2012; Croft et al., 2016). We note that ground based observations are performed under clear sky conditions, while the considered collocated GC grid cell can be partially cloudy. This effect could be mitigated by using the aforementioned fine grid high resolution simulations in the future.

One way to qualitatively assess the quality is to break down the individual components into those that contribute mainly to the FM or those that contribute more to the CM fraction. Using AERONET SDA retrievals of the CM and FM fraction we compare the GC FM and CM fractions with those of AERONET. Fig. 4 shows the seasonal AODs for spring and summer, respectively, averaged over the full period. Over each station, three circles are shown and display (i) the proportion of FM (red) and CM (blue) according to AERONET; (ii) the same as (i) but according to GC; (iii) the corresponding full AOD component speciation according to GC.

The CM AOD shows variation with 11% and 7% between the sites, located over CA and over Hornsund during MAM and JJA respectively. According to the GC speciation pie-chart the difference in CM AOD between these places is driven by the abundance of sea salt (around 9%) during JJA. GC AOD overestimates AERONET FM AOD during spring while producing comparable results during summer. The climatological difference in the FM and CM AOD between AERONET and GC might be traced back to a potential overestimation of haze events of GC during spring.

Using corresponding atmospheric profiles and zonally averaged contour plots of dust aerosol illustrate that over all AERONET sites elevated dust layers during spring are prevalent, whereas such long-range transport is weaker during summer. This finding is in agreement with that from (Breider et al., 2014; Stone et al., 2014).

During both seasons according to the GC simulations, the strongest contributor to FM AOD is sulfate aerosol. Compared to spring the sulfate decreases by 3.3% at CA sites and 4.1% at Spitsbergen during summer (Table.1). In general, compared to spring, the summer season is dominated by FM AOD, which is observed and predicted by both AERONET and GC simulations, in agreement with (Willis et al., 2018).

We also note that organic carbon (OC) is increasing by 6% and 1.5% over all sites during summer compared to spring (Table.1). We suspect that the increase in OC and the larger presence of FM AOD at PEARL, OPAL, and Thule indicate the impact by boreal forest fires. This finding agrees with that from (Sand et al., 2017; Xian et al., 2022). The evidence of boreal forest fires (BC+OC) is also seen in the contour maps in Fig. A5.

In summary, the validation and evaluation statistics (R, rmse) shown in Fig. 2, indicate a reasonable agreement between all the three datasets. The seasonal climatology shown in Fig. 3, exhibit a spring time maximum, which is consistent with the enhanced transport to the Arctic during this time of the year. Overall, sulfate, partly mineral dust and carbonaceous aerosols are the dominating contributors during spring and summer according to GC AOD. The AOD from GC and AEROSNOW in comparison to AERONET showed reasonable to good agreement over the four stations in the high Arctic.



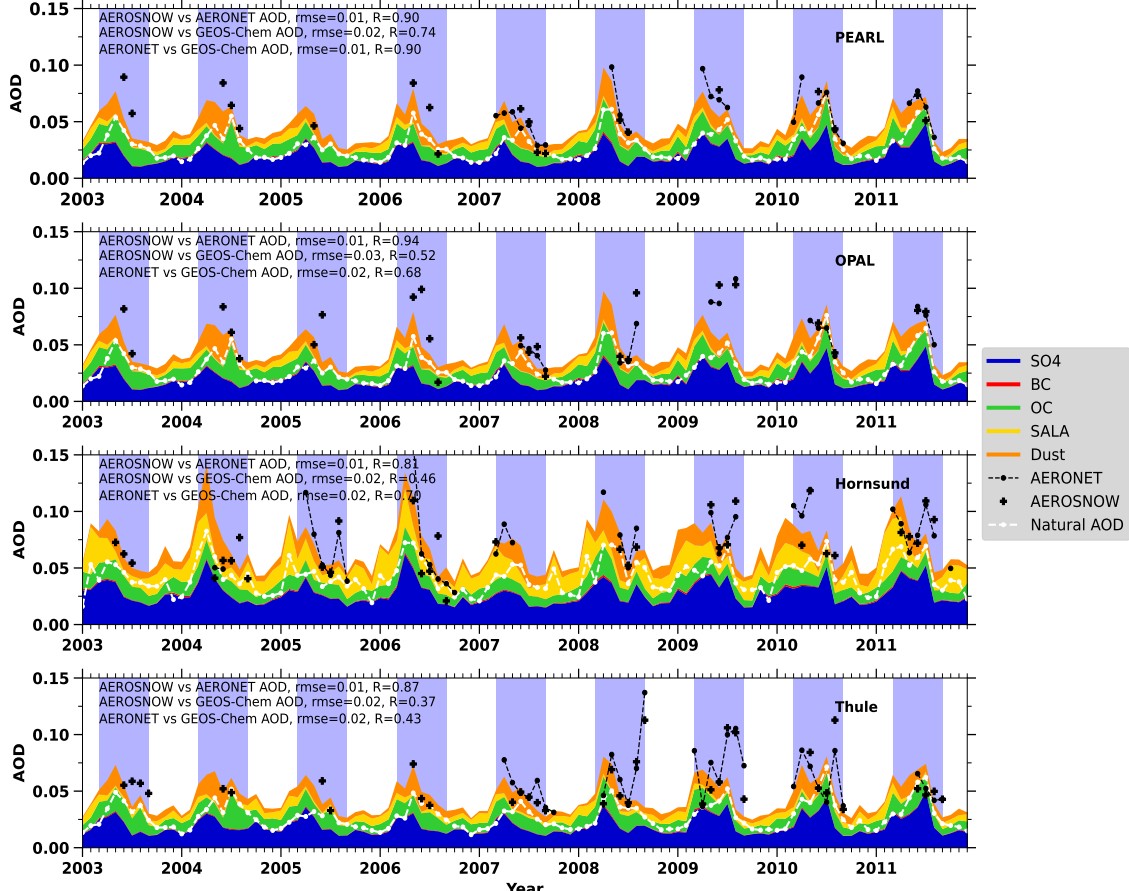

**Figure 2.** Monthly mean time-series of GC AOD, speciated and natural, and the AERONET and AEROSNOW measured AOD at PEARL, OPAL, Hornsund and Thule stations. The MAM,JJA periods are highlighted with blue shades. Annotations for each time series show rmse and R between AEROSNOW and AERONET AODs.

## 4 Results: Arctic AOD climatology and boreal forest fires

In the above comparisons of AEROSNOW and GC AOD with AERONET AOD have enalbed us to asses their quality. In the following we invesitgate the AOD measured and simulated in regions covered by snow and ice beyond the AERONET measurment sites.

We have created and analyzed seasonal climatologies of the AOD during Arctic spring and summer over sea ice, derived from space-based retrievals and GC simulations.

The 1990s were dominated by a decline in AOD (see Schmale et al. (2022)). This is attributed to (i) the reduction in industrial activity and the related release of pollution in the former Soviet Union countries, and thus a substantial loss of long-





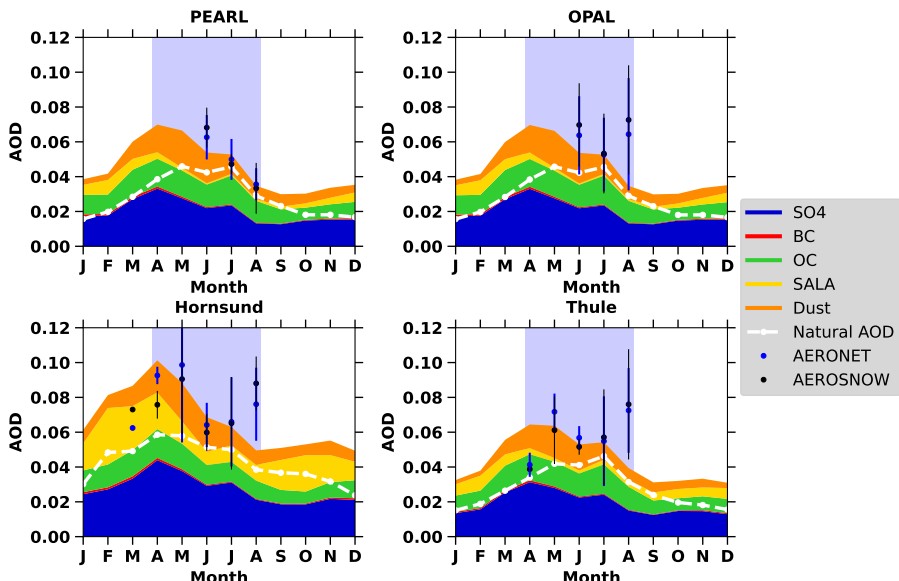

**Figure 3.** Seasonal AOD variation over PEARL, OPAL, Hornsund and Thule with AERONET and AEROSNOW values average from 2003 to 2011. Blue and black circles denote observed monthly mean AODs, and vertical error bars show 1 standard deviation of the means for AEROSNOW and AERONET respectively. Stacked contours show the speciated AOD contribution from GC simulations. AOD from natural sources is shown as a white dashed line.

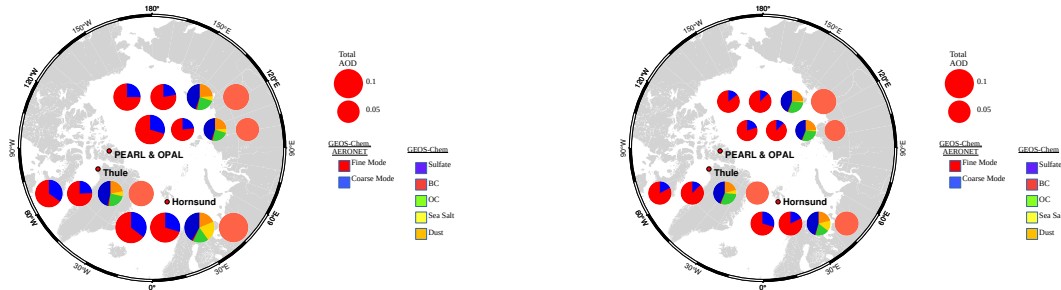

**Figure 4.** Arctic polar map with red dots depicting the locations of the AERONET stations. Left panel: MAM; Right panel: JJA. The circles from left to right of each panel shows show, FM and CM AODs from AERONET, FM and CM AODs from GC, the speciated pie-charts AODs from GC, and AEROSNOW retrieved AOD for each stations. Red colors represent fine mode and blue colors represent coarse mode.



range transport of pollutant aerosols and precursors, and (ii) effective European and North American air quality legislation, which also contributed to reduced pollutant transport to the Arctic.

The beginning of the 2000's (the study period), however, is potentially a turning point. For example, there may be increasing local sources of AOD from the decreasing sea ice extent or an increase in sub-Arctic forest fires. We, thus, derive the percentage

contribution of component AOD to identify potential changes and relate them to Arctic boreal forest fires.

## 4.1 Spring and Summertime AOD Climatology over the Arctic Sea Ice

Similar to the analysis shown in Fig. 2, where we examined the time series of total and component-based AOD from AEROS-NOW and GC over AERONET stations, we now discuss how AOD values change over time across the entire region of Arctic sea ice.

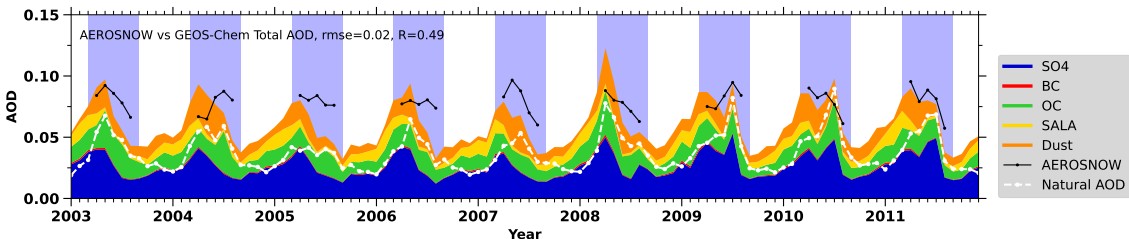

**Figure 5.** Monthly mean time-series of GC speciated and Natural AOD, and AEROSNOW AOD over Arctic sea-ice. The MAM,JJA periods are highlighted with blue shades. Annotations for each time series show rmse and R between different AODs.

When evaluating the AOD datasets over pan-Arctic sea ice, depicted in Fig. 5, we observe comparable GC and AEROSNOW AOD, having an R value of 0.49 and rmse value of 0.02 (Fig. A4). The GC AOD exhibit higher values in spring 2009 and lower ones during spring 2007 while showing comparable AOD during the Arctic summer. However, by comparing seasonal climatology averaged from 2003 to 2011, GC results and AEROSNOW show partly good agreement during spring whereas during summer GC AOD is lower than AEROSNOW AOD (Fig. 6). We propose that these differences result from the excessive

precipitation in GC. This leads to increased wet scavenging (precipitation averages for the Arctic are also shown in Fig. A7. Alternatively, the secondary aerosol formation in the model may be insufficient. Frequent new particle formation over the high Arctic pack ice by enhanced iodine emissions may occur (Baccarini et al., 2020). On the other hand, the AEROSNOW retrievals could also be affected by cloud contamination. High levels of cloud cover are observed over the Arctic in summer with average values around 0.8 (Kato et al., 2006), and although we adopted reasonable cloud masking for the AOD retrievals,

we cannot exclude completely the possibility that residual cloud contamination may have an impact (Jafariserajehlou et al., 2019).

Additionally, both AEROSNOW and GC capture the higher values of AOD over teh north of Alaska and Siberia during summer, a region which is often influenced by boreal forest fires during the study period. Overall, GC AOD appears to agree with AEROSNOW AOD over central Arctic sea ice. This implies that the component AOD in this GC model is realistic.



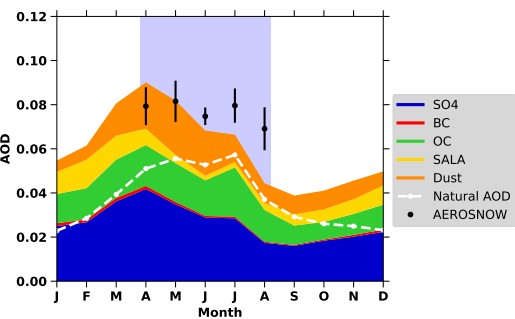

**Figure 6.** Seasonal AOD variation over Arctic sea-ice with AEROSNOW values average from 2003 to 2011. Black circles denote observed monthly mean AODs, and vertical bars show 1 standard deviation of the means for AEROSNOW. Stacked contours show the speciated AOD contribution from GC simulations. AOD from natural sources is shown as a white dashed line.

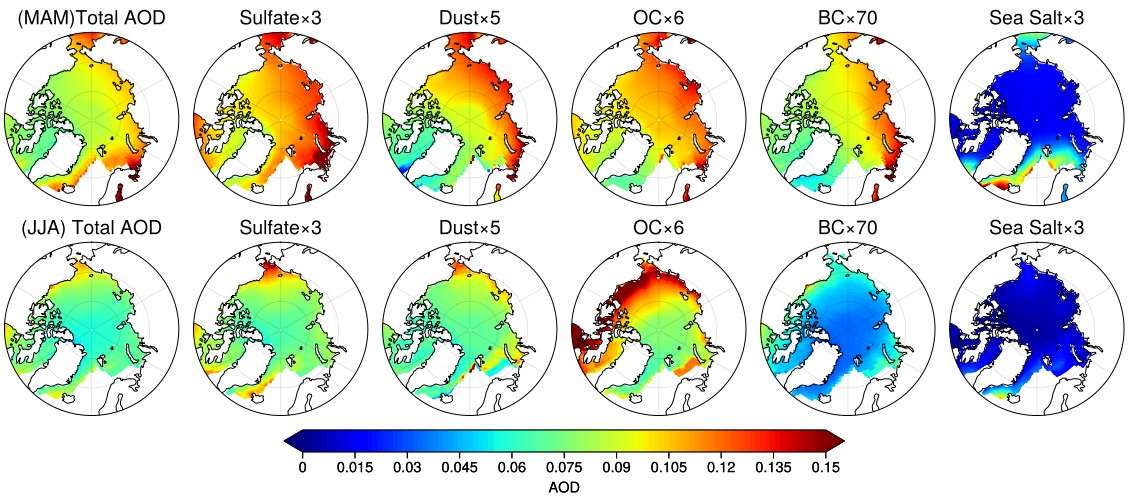

**Figure 7.** Mean climatological MAM (top panel) and JJA (lower panel) GC simulated total and speciated AOD over Arctic Sea Ice averaged from the year 2003 to 2011.





The spring and summer time spatial distribution of monthly mean total AOD and component AOD over Arctic sea ice averaged from 2003 to 2011 is shown in Fig. 7 as the first and second row respectively.

During spring the higher values of AOD (0.1-0.12) are observed near European and the Asian continents and smaller values (0.07-0.08) towards CA and Greenland. Spring values are mostly dominated by long range transport of aerosols formed as a result of anthropogenic activity at lower latitudes in Europe, America and Asia. This is also observed in the zonally averaged

contour plots in Fig. A5 and also by Stone et al. (2014). Fig. A6 shows the transport features represented by vertical AOD accumulation between 600hPa and 300hPa. In Fig. 5, the spring maxima during 2003, 2006, 2008 are probably associated with the transport of wide-spread agricultural burning in the high latitudes (Saha et al., 2010; Stohl et al., 2006).

The component AODs show stronger variability in spring compared to summer due to the different sources of aerosol, explained above, in the different seasons. The FM AOD is dominant in both spring (67%) and summer (72%) but has a higher

relative contribution during spring (Fig. 8). The contribution of the sulfate to the total AOD over Arctic sea ice is decreased by 3.0%, while carbonaceous aerosols have increased by 8.4% during summer compared to spring, averaged during the study period (Fig. 8; Table.1). The increase of BC+OC during summer, when long-range transport from the mid-latitudes is not significant, confirms the relevance and penetration of Arctic boreal forest fires deep into the high Arctic sea ice covered areas (Fig. 10). The black box in Fig. A5 shows the latitudinal range in which forest fire originate.

According to GC (as depicted in Fig. 7) sea salt, as a result of sea spray, is prominent over the Greenland Sea, Norwegian Sea, North Atlantic, and the Bering Strait (North Pacific). This is explained by the high wind speed, especially in spring. The maximum values of AOD (0.09-0.08) over sea ice is in the months of April and May, whereas the minimum occurs during July and August, and September. The latter is most likely explained by elevated precipitation and wet scavenging in GC Fig. A7.

Fig. 9 shows the zonal monthly mean of AOD component variability averaged over the period of 2003-2011. We attribute the

lower AOD to higher rates of aerosol removal during summer over Arctic sea ice. The zonal average of AOD from 60N to 90N over sea ice, averaged for the study period, shows that the AOD values are high at 60N and decrease gradually with increasing latitude (Fig. 9). However, the OC+BC AOD has a peak during summer, whereas all the other aerosol components decrease. We suspect that this is again due to wet scavenging: in GC 50% of OC emitted from all primary sources are hydrophobic (Cooke et al., 1999; Chin et al., 2002). The combination of hydrophobicity and increasing boreal forest fires makes carbonaceous

aerosols (BC and OC) a potential large contributor to the total AOD over Arctic sea ice during summer.

## 5   Conclusions

This study investigated the spatio-temporal variability and seasonality of Arctic optical aerosol depth (AOD) and AOD components over the cryosphere by using (i) the AEROSNOW retrieval algorithm (Swain et al., 2023a) applied to nine years of AATSR space data; (ii) AERONET ground based AOD observations; and (iii) GC model simulations of AOD during the period

from 2003 to 2011.

  i)  **AOD retrieval over Arctic snow and ice:** AOD has been determined over the pan-Arctic snow and ice surface over a
       nine-year period and used to evaluate global atmospheric 3-D chemical transport using GC. The retrieval of AOD over



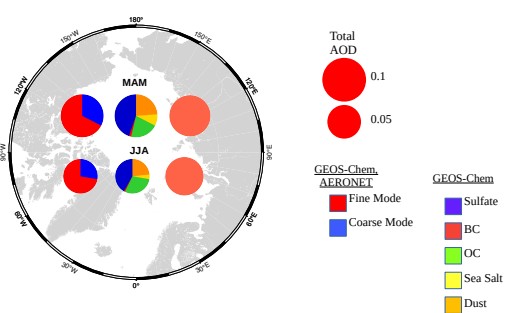

**Figure 8.** Arctic polar map showing pie-charts from left to right: FM (red) and CM (blue) AOD, the speciated AODs from GC model and AEROSNOW retrieved AOD over Arctic sea ice for JJA and MAM respectively.

| Locations | Latitude | Longitude | Elevation (m) | Region | BC (JJA%-MAM%) | OC (JJA%-MAM%) | Dust (JJA%-MAM%) | SALA (JJA%-MAM%) | SO4 (JJA%-MAM%) |
|---|---|---|---|---|---|---|---|---|---|
| PEARL | 80.054N | 86.417W | 615 | Arctic Archipelago | -0.66 | 6.21 | 1.70 | -3.89 | -3.36 |
| OPAL | 79.990N | 85.939W | 0 | Arctic Archipelago | -0.66 | 6.21 | 1.68 | -3.88 | -3.35 |
| Hornsund | 77.001N | 15.540E | 10 | Svalbard | -0.46 | 1.58 | 2.52 | -6.99 | 3.34 |
| Thule | 76.516N | 68.769W | 225 | Arctic Archipelago | -0.62 | 5.93 | -0.44 | -0.75 | -4.12 |
| Sea Ice | 60N to 90N | 180W to 180E | 0 | High Arctic | -0.53 | 8.36 | -0.56 | -4.31 | -2.95 |

**Table 1.** Difference of the percentage of speciated AOD between MAM and JJA shown in Fig. 4, Fig. 8.

Arctic snow and ice using AEROSNOW algorithm (Swain et al., 2023a) and AATSR observations is able to provide AOD otherwise unavailable in the MODIS data products over the cryosphere.

The high anthropogenic aerosol loading (Arctic Haze events) due to long-range transport over Arctic snow and ice is captured by the AOD behavior in the AEROSNOW AOD as well as that of GC. The time series and seasonality of the GC, AEROSNOW, and AERONET AOD agree well. A partial overestimation of AEROSNOW is observed over Arctic sea ice in 2005 and 2007, which may be due to uncertainties in surface parameterization and aerosol types in this region. Further improvement of the AOD retrieval could be possible in terms of improved knowledge of cloud masking,
surface reflectivity properties, and the adoption of more appropriate aerosol types. The promising results obtained with





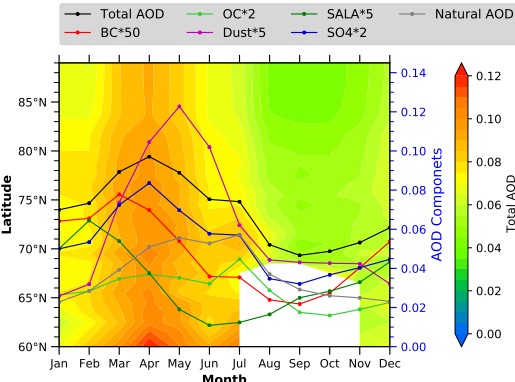

**Figure 9.** Zonal averages of total AOD over Arctic sea-ice as a function of month and latitude for GC model, superimposed with climatological (2003-2011) seasonal cycle of total and speciated AOD over Arctic sea-ice. The total AOD is monthly averaged in the period 2003-2011. The white space shows the receding of sea ice from 60N to 70N over the Arctic in summer.

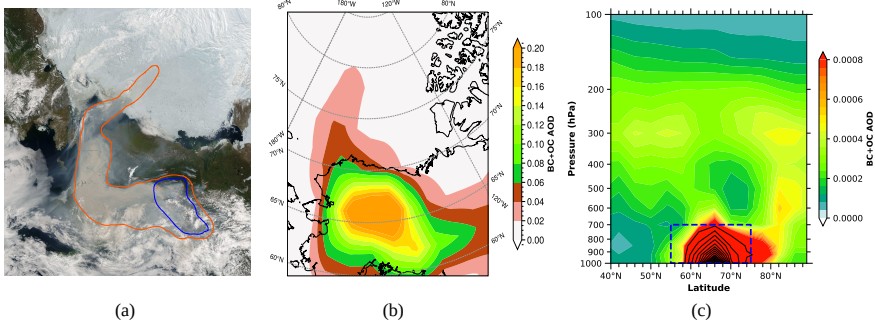

(a)                               (b)                               (c)

**Figure 10.** An example of boreal forest fire smoke intrusion into the high Arctic from fires originated in Alaska. a) True-color Terra satellite imagery taken on 1st July 2004. Red dots within the blue contour shows satellite-detected fire hotspots. b) GC BC+OC AOD simulated for 1st July 2004. c) GC vertical distribution showing the BC+OC layers around the source area.

AEROSNOW are suitable for use in a) the evaluation and improvement of AOD simulated by chemical transport models (CTMs) and b) the determination of climatological analyses, over the homogeneous Arctic sea ice regions, vulnerable to climate change.

**ii) Arctic AOD climatology:**

Over Arctic snow and ice, especially sea ice, the AEROSNOW AOD and the GC model simulations of AOD show similar spatio-temporal amounts and variations. Fine mode aerosols dominate in both spring and summer, and have a higher contribution in summer compared to spring. The contribution of anthropogenic aerosols to total AOD is dominant in spring, while naturally occurring aerosols predominate in summer. The percentage contribution of carbonaceous aerosol (BC and OC) to the total AOD is higher in summer than in spring at all AERONET sites and on sea ice. The fraction





of sulfate and dust is slightly higher in spring than in summer. The zonal variation in AOD largely extends from 60°N latitude to 90°N latitude in spring, which is due to the high precipitation that leads to wet deposition over the Arctic in summer (Fig. 9), which agrees with (Garrett et al., 2011).

    Although precipitation and wet deposition are high, primary carbonaceous aerosols, particularly organic carbon (OC), peak in summer because they are more hydrophobic. The combination of hydrophobicity and increasing boreal forest

fires means that carbonaceous aerosols (black carbon, BC, and OC) are an increasingly important contributor to total AOD over Arctic sea ice in summer (Cite some articles here). BC AODs are conspicuous in both seasons, but during the spring (MAM) they result from long-range transport of anthropogenic pollution. This is not the case in summer (JJA).

    In addition, the relative percentage contributions of sulfate to total AOD decreases over Arctic sea ice, whereas carbonaceous aerosols increase more in summer than in spring, averaged from 2003 to 2011.

**iii) Overall performance of the AEROSNOW retrieval and GEOS-Chem model simulations:**

    Both the AEROSNOW AOD and GC model simulations of AOD show good agreement with the ground-based AERONET measurements over snow and ice and were used for further analysis of their variability and climatology. The GC simulations perform better for FM and CM AOD in summer than in spring over the AERONET sites. An advanced aerosol retrieval algorithm, AEROSNOW, was used to retrieve AOD over Arctic snow and ice. This provides a potentially new

dataset of AOD over the Arctic. We conclude that improved meteorology and emission inventories for central Arctic sea ice during spring and summer will improve the accuracy of the GC AOD. This will in turn facilitate the assessment of the AA and the related sea ice loss.

    The AOD obtained from AEROSNOW can be effectively used as a baseline for various models, especially over the Arctic sea ice in spring and summer for the period 2003-2011, which is the period when the Arctic amplification was first clearly

identified. We propose the use of the AEROSNOW datasets to assess CTMs and constrain climate models which simulate the direct and indirect effects of aerosols on AA. The observation of a summer peak indicated that natural sources of AOD are changing.





## Appendix A: Additional Figures

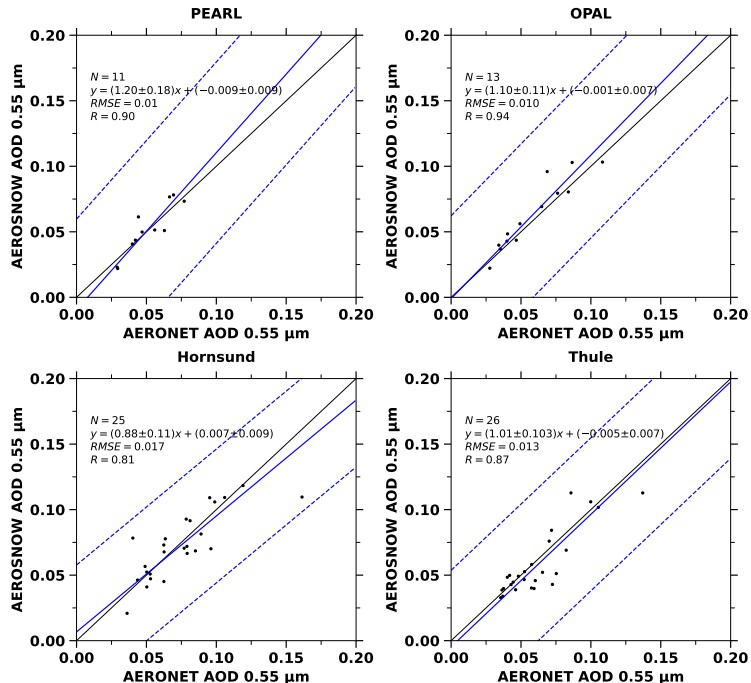

**Figure A1.** Validation of monthly mean AEROSNOW retrieved AOD colocated with monthly mean AERONET observation AOD obtained over PEARL, OPAL, Hornsund, and Thule stations. The linear regression lines are shown as blue dashed line.



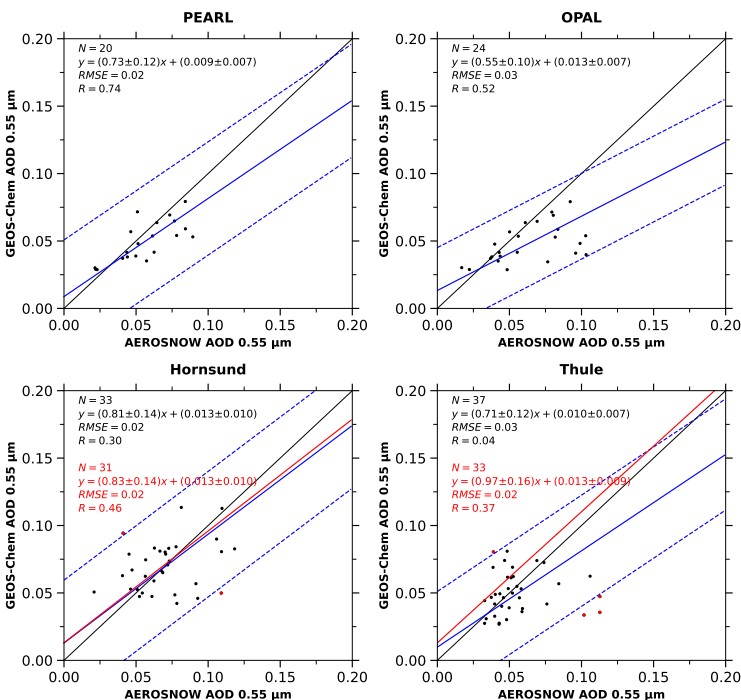

**Figure A2.** Validation of monthly mean GC AOD with monthly mean AEROSNOW observation AOD obtained over PEARL, OPAL, Hornsund, and Thule stations. The linear regression and one-standard deviation lines are shown as a solid blue and dashed blue lines. Further, red regression line is without the out layers marked as red





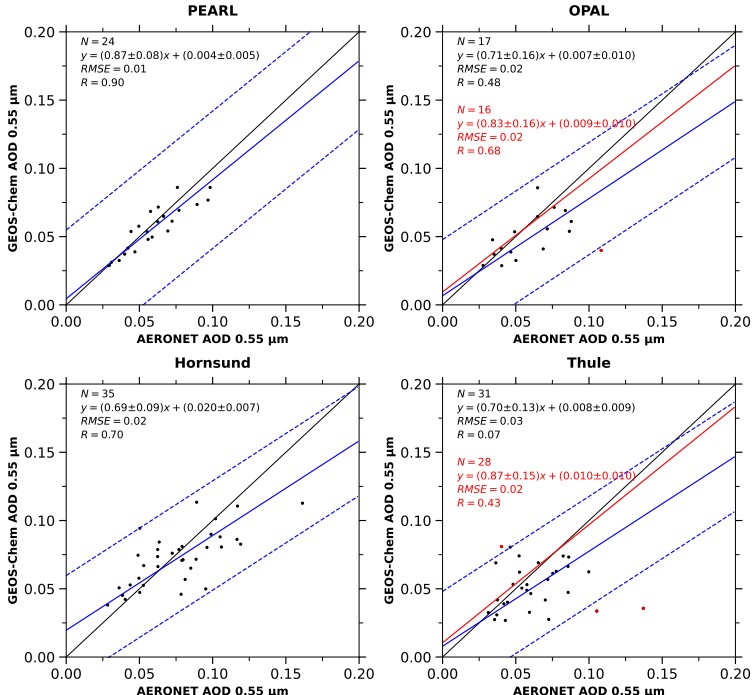

**Figure A3.** Validation of monthly mean GC AOD with monthly mean AERONET observation AOD obtained over PEARL, OPAL, Hornsund, and Thule stations. The linear regression and one-standard deviation lines are shown as a solid blue and dashed blue lines. Further, red regression line is without the out layers marked as red

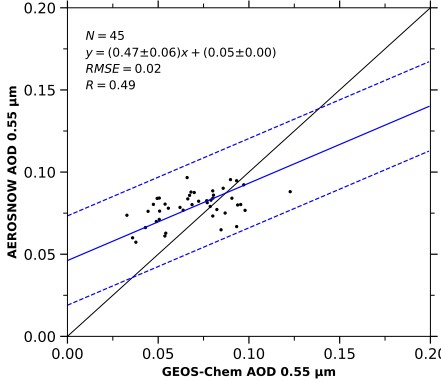

**Figure A4.** Validation of monthly mean GC AOD with monthly mean AEROSNOW observation AOD obtained over vast Arctic Sea-Ice. The linear regression and one-standard deviation lines are shown as a solid blue and dashed blue lines.





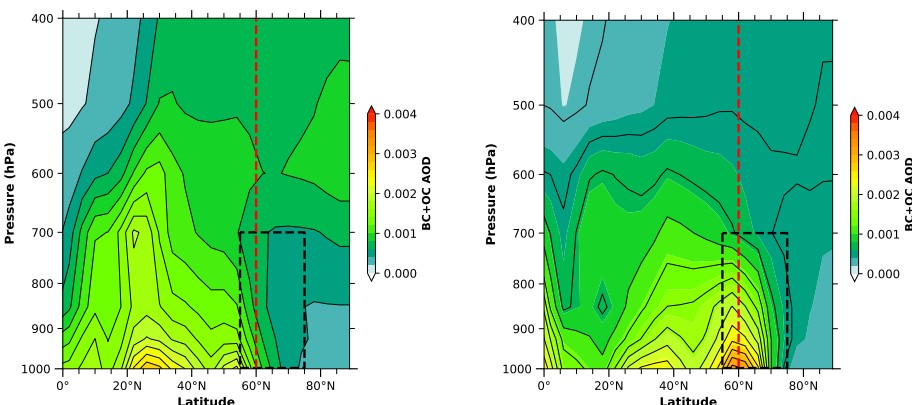

**Figure A5.** Vertical zonal mean of total, BC+OC AOD over PEARL, 0PAL, Hornsund and Thule for (a) MAM and (b) JJA respectively, averaged from the year 2003 to 2011. Black box shows the biomass burning within Arctic.

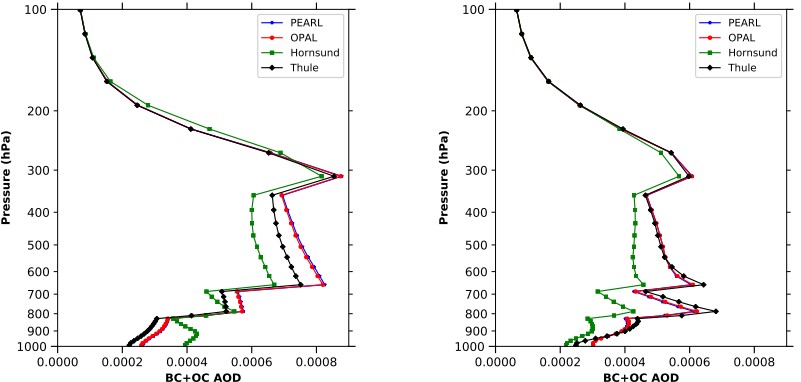

**Figure A6.** Vertical distribution of total, BC+OC AOD over PEARL, 0PAL, Hornsund and Thule for (a)MAM and (b) JJA respectively, averaged from the year 2003 to 2011.



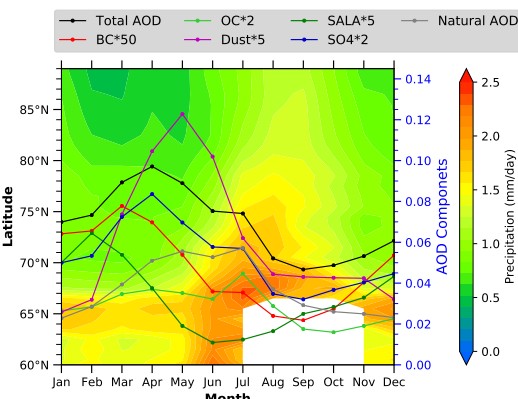

**Figure A7.** Zonal averages of total precipitation over Arctic sea-ice as a function of month and latitude for GEOS-Chem model, superimposed with climatological (2003-2011) seasonal cycle of total and component AOD over Arctic sea-ice. The total precipitation is monthly averaged in the period 2003-2011. The white space shows the receding of sea ice from 60N to 70N over Arctic in summer.



*Code and data availability.* The code and data supporting the conclusions of this manuscript are available upon request.

*Author contributions.* B.S., M.V. conceived the research. B.S. has done GEOS-Chem model setup, simulations and processed the aerosol data, analyzed all records and wrote the manuscript. M.V., A.D., L.L., Y.Z., A.S, S.S.G., J.P.B. helped in shaping this manuscript. Funding acquisition by M.V. and J.P.B. All authors contributed to the interpretation of the results and the final drafting of the paper.

*Competing interests.* The authors declare that they have no conflict of interest.

*Acknowledgements.* We thank GEOS-Chem model community for making the data available, ESA for AATSR data set. This work has been
funded by the Deutsche Forschungsgemeinschaft (DFG, German Research Foundation) within the project "ArctiC Amplification: Climate Relevant Atmospheric and SurfaCe Processes, and Feedback Mechanisms (AC)[3]" as Transregional Collaborative Research Center (TRR) 172, Project-ID 268020496.



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
