# Peer review of "Spring and summertime aerosol optical depth variability over Arctic cryosphere from space-borne observations and model simulation"

_EGUsphere, 2023_

## Author Comment (AC1)

**Referee #1**

**Previous title:** Spring and summertime aerosol optical depth variability over Arctic cryosphere from space-borne observations and model simulation

**Revised title:** Variability of aerosol over the Arctic cryosphere from space-borne observations and model simulations for the cycle of sea ice growth and decline.

The authors thank the referee for her/his effort, and time taken to review our manuscript. The valuable criticisms and comments have helped us to improve our paper. We hope that we have been able to answer satisfactorily the questions raised and clarify parts of the manuscript which were unclear or ambiguous.

We have changed the title of the manuscript following the recommendation of referee.

In the following the referee comments and criticisms, our responses, as authors, and our resultant changes to the manuscript are colored **black**, **blue** and **red** respectively.

**Spring and summertime aerosol optical depth variability over Arctic cryosphere from space-borne observations and model simulation**

Swain et al.

**Summary:**

This work builds on AERONET data and a satellite monthly AOD product for the Arctic (AEROSNOW) described in a separate paper, to evaluate the performance of GEOS-Chem, and describe the seasonality of AOD per species and mode for 4 stations in the Arctic and for the sea-ice regions. The main conclusions are that GEOS-Chem reproduces adequately the known features of Arctic aerosols (haze, intrusions of biomass burning plumes…). Some suggestions are proposed to explain the observed discrepancies between GEOS-Chem and AERONET/AEROSNOW.

**General comments**

The idea of this work is of interest for the community, as a better knowledge and understanding of Arctic aerosols is a critical topic. The manuscript is generally well written, and most of the figures are easy to read.

**Q1:** My biggest concern is that the study relies on a methodology for satellite AOD presented in a companion paper that is under review in AMT: *Swain et al., 2023a. Spring and summertime aerosol*

*optical depth retrieval over the Arctic cryosphere by using satellite observations*. (same authors, almost same title, several common figures).

This is problematic for 2 reasons:

**a)** I cannot review the other paper and assess its validity, so the AEROSNOW product is not officially validated until this is published, and therefore not really fit for use here.

**Response:** It is true that the methodology for the AEROSNOW approach has been submitted to AMT in a separate paper at this time. However, the first version of AEROSNOW has already been presented and validated by Istomina et al., (2009) and Mei et al., (2013), as we also presented in the manuscript at line 67-68. Recent improvements to the approach of AEROSNOW made it possible for the first time to run the approach over the entire central Arctic cryosphere with high spatio-temporal coverage. For this reason, we believe it is important to publish the improved approach and its validation in AMT. The manuscript is publicly available in AMT-D and can be found here together with the referees' comments and our rejoinder: https://amt.copernicus.org/preprints/amt-2023-65/.

We therefore see no fundamental problems in the availability of the information for the referees and the editor in this journal and would like to ask referee to view the information there accordingly. Based on the validations performed by Istomina et al., (2009,2011) and Mei et al., (2013) and ours (presented in AMT-D), we conclude that the AEROSNOW dataset is well suited for the present study.

**b)** AEROSNOW is presented here as one of the main value-added of the paper, but it is already going to be published in the companion paper, including with similar/same plots in both manuscripts. All of the AERONET and AEROSNOW-based analyses and results shown here are already presented in the other paper. Therefore, the only visible difference is the GEOS-Chem analysis. At the very least, the scope of the paper should be changed, explicitly stating that the new results are only from the GEOS-Chem modeling and focusing on this part.

**Response:** [The answer to this question is partly described in the author's response to the question Q2]

We would like to cite the two figures used in this ACP manuscript from our other paper that is under review in AMT: "*Swain et al., 2023. Spring and summertime aerosol optical depth retrieval over the Arctic cryosphere by using satellite observations, doi: https://doi.org/10.5194/amt-2023-65*".

These two figures are as follows:

i) *Figure.1,* with title "Location of PEARL(80.054°N, 86.417°W), OPAL(79.990°N, 85.939°W), Hornsund (77.001°N, 15.540°E) and Thule(76.516°N, 68.769°W) AERONET measurement stations considered in this study".

ii) *Appendix Figure A1*. With title *"Validation of monthly mean AEROSNOW retrieved AOD colocated with monthly mean AERONET observation AOD obtained over PEARL, OPAL, Hornsund, and Thule stations. The linear regression lines are shown as blue dashed line"*.

We would like to point out to the reviewer that the largest uncertainties result from the direct and indirect effects of aerosols on clouds and climate in the Arctic, which are poorly

represented in models (Boucher et al., 2013). This deficiency is particularly true for the central Arctic sea ice region due to the lack of datasets with high spatiotemporal observational coverage and understanding of aerosols in this region (Schmale et al., 2021). It is worth noting that several valuable studies have been conducted for Arctic aerosols using models (such as, Hardenberg et. al., 2012; Evangeliou et. al., 2016; Sand et. al., 2017; Breider et. al., 2017; Ren et. al, 2020; Schmale et. al., 2021), reanalysis datasets ( such as, Xian et. al., 2021; Xian et. al., 2022 ), and ground-based observations (such as, Tomasi et al., 2015; Schmale et. al., 2022; Creamean et. al., 2022; Schmale et. al., 2022). All of these important and valuable studies do not have aerosol observations over the entire Central Arctic cryosphere with high spatial and temporal coverage and urged advanced AOD retrieval algorithms from the space-based sensors for the central Arctic cryosphere. We therefore think, that our study will help to improve aerosol model evaluations and better constrain the radiative and potentially indirect effects of aerosol to assess the impact of aerosol on Arctic warming (Boucher et. al., 2013; Schmale et. al., 2021).

In this study, we have divided the objective of this manuscript as follows:

> i) To demonstrate how the total AOD first obtained by AEROSNOW can serve as a baseline for constraining chemical models over this region by evaluating the GEOS-Chem 3D chemical transport model to address the lack of aerosol observational data over the rapidly changing, sensitive central Arctic cryosphere. In addition, this dataset can serve as a baseline for testing several other models over this subregion of the Arctic.

> ii) We not only evaluate the models against space and ground-based measurements but also performed a large-scale analysis focusing on the more spatio-temporal coverage with regional, seasonal and annual variability of total AOD and its components originating from anthropogenic as well as natural  sources with impact of smoke intrusion events, seasonal change in precipitation over the central Arctic cryosphere. This will further confirm mechanisms such as precipitation and smoke intrusion to the central Arctic cryosphere that is driving the total aerosol distribution retrieved by AEROSNOW for the entire cycle of sea-ice growth and decline seasons as well as annual variations for the period nearly a decade (from year 2003-2011).

We have expressed these objectives in our manuscript as follows:

The _first objective_ is defined at line 78-81 as *"The scientific objective of this study is to use satellite observations of total AOD and analyze the AOD components, transport, meteorological conditions, and natural and anthropogenic aerosol sources that   determine total AOD over the high Arctic cryospheric regions. At the same time, model results were compared with AEROSNOW results in the absence of in situ and ground- based observations of AOD over high Arctic sea ice.".*

The _second objective_ is defined at line 94-96 as *"Consequently, we have  extended our objective to examine aerosol composition over the snow- and ice-covered regions of the high Arctic (72-90°N, as defined by Sand et al. (2017)). To achieve this goal, we investigate aerosol composition in the high Arctic using GEOS-Chem, a global chemical transport model (CTM) simulation.".*

Consequently, we believe, that the scope is well defined and in line with the narrative of the manuscript.

We suggest to make the following changes to the introduction to highlight the necessity and novelty of our study.
**We propose to add a citation to title of Figure 1 and Appendix Figure A1 :**
Figure 1. Location of PEARL(80.054°N, 86.417°W), OPAL(79.990°N, 85.939°W), Hornsund(77.001°N, 15.540°E) and Thule(76.516°N, 68.769°W) AERONET measurement stations considered in this study (Swain et al., 2023).

Figure A1. Validation of monthly mean AEROSNOW retrieved AOD colocated with monthly mean AERONET observation AOD obtained over PEARL, OPAL, Hornsund, and Thule stations. The linear regression lines are shown as blue dashed lines (Swain et al., 2023).

**At line 74-77, we propose to modify:**
Furthermore, in addition to ground-based measurements and satellite observations, several valuable studies using models (such as Hardenberg et. al., 2012; Evangeliou et. al., 2016; Sand et. al., 2017; Breider et. al., 2017; Ren et. al, 2020; Schmale et. al., 2021) and reanalysis datasets (such as, Kinne et. al., 2019; Xian et. al., 2021; Xian et. al., 2022) have been conducted over the Arctic. It is worth noting that all of these studies lack observational data, especially over the high Arctic cryosphere region, which is highly vulnerable to warming and climate change in the Arctic (Bintanja et. al., 2017).

**At line 94-96, we propose to modify:**
Consequently, we have extended our objective to examine aerosol composition over the snow- and ice-covered regions of the high Arctic (72-90°N, as defined by Sand et. al.,2017) originating from anthropogenic as well as natural sources with impact of smoke intrusion events and seasonal change in precipitation over the central Arctic cryosphere. To achieve this goal, we investigate aerosol composition in the high Arctic using GEOS-Chem, a global chemical transport model (CTM) coupled with MERRA-2 (Molod et al., 2015) meteorology simulation.

**Q2:** As a result, this paper is mostly a model evaluation of Arctic AOD simulated by GEOS-Chem, albeit with a new data set for evaluation, showing that the model behaves as expected compared to what was already known in the literature. Therefore the conclusions do not really bring new knowledge. In particular the seasonality and composition of Arctic aerosols, including over sea ice, which is the focus here, is already described in recent literature such as Sand et al., 2017; Schmale et al., 2022; Moschos et al., 2022. In particular, the AEROCOM models evaluated for the Arctic in Sand et al., 2017 include GEOS-Chem.
**Response:**
The scientific significance of this manuscript is described in the author's response to question Q1(b).

We are well aware of the referee's suggested articles on this topic. We would like to stress that the articles proposed by Referee (Sand et al., 2017; Schmale et al., 2022; Moschos et al., 2022) point out in their own articles the lack of spatiotemporal observational data on aerosol loading over the central Arctic cryosphere region. There, they emphasize the need to develop new sophisticated aerosol data sets that can be used to assess and constrain their own studies, particularly over this subregion of the Arctic.

To avoid misunderstanding we would like to briefly summarize the way in which we classify the articles mentioned by the referee (Sand et al., 2017; Schmale et al., 2022;  Moschos et al., 2022) as follows:

i.   *Sand et. al., 2017. Aerosols at the poles: an AeroCom Phase II multi-model evaluation. Atmos Chem Phys. 17, 12197-12218. doi: 10.5194/acp-17-12197-2017*

In Sand et. al., 2017 the authors have used MODIS and CALIOP for the evaluation of 16 different AeroCom Phase II models over Arctic and Antarctica.

At page number 12203 and paragraph 3 of the Sand et. al., 2017, the authors particularly mentioned that *"Retrievals of AOD from the MODIS satellite directly over snow and sea-ice are not available due to the high reflectivity of these surfaces. Glantz et al. (2014) have provided spatial averages of MODIS AOD 555 nm over (darker) ocean"* and further, *"CALIOP has an inclination angle of about 98.14° and therefore has no data points over central Arctic cryosphere"*. Thus, the article is obviously emphasizing the lack of observational data over central Arctic cryosphere. We do not directly quote this sentence in our article but state in line 57 of our manuscript as follows  "These sparse AOD measurements are used in part to explain AOD variations in climate models (Sand et al., 2017; Palazzi et al., 2019)."

In addition, it is important to note that the study conducted in Sand et al. (2017) using AeroCom models, which include GEOS-Chem (as mentioned by the referee), ran simulations for only one year (that is the year of 2006, with emissions for 2000).  In our study, on the other hand, we used simulations for nearly decade (from 2003 to 2011).

ii.   *Schmale et al., 2022. Pan-Arctic seasonal cycles and long-term trends of aerosol properties from 10 observatories. Atmos Chem Phys. 22, 3067-3096. doi:10.5194/acp-22-3067-2022*

In Schmale et al. (2022), the authors analyzed 9 chemical aerosol species and 4 optical particle properties from 10 Arctic observatories (Alert, Kevo, Pallas, Summit, Thule, Tiksi, Barrow/Utqiagvik, Villum, and Gruvebadet and Zeppelin Observatory-both at Ny-Ålesund Research Station) to understand changes in anthropogenic and natural aerosol contributions. Variables include equivalent black carbon, particulate sulfate, nitrate, ammonium, methanesulfonic acid, sodium, iron, calcium, and potassium, as well as scattering and absorption coefficients, simple scattering albedo, and scattering angstrom exponent.

It is worth noting that, this study is an important one but the observations over ten Arctic stations do not necessarily cover (spatio-temporally) the vast central Arctic cryospheric region.

iii.   *Moschos et al., 2022. Elucidating the present-day chemical composition, seasonality and source regions of climate-relevant aerosols across the Arctic land surface. Env Res Lett. 17, 034032. doi: 10.1088/1748-9326/ac444b*

In Moschos et. al., 2022, the authors have analyzed dataset on the overall chemical composition and seasonal variability of the *Arctic total particulate matter* (with a size cut at 10 µm, or without any size cut) at eight observatories.

As the title of Moschos et. al., (2022) suggests, the analysis of particulate matter was performed only over Arctic land areas. The eight observatories do not cover the vast central Arctic cryosphere. Further, this study is limited to particulate matter only.

Therefore, we do not agree with the referee that central parts of our study have already been published in the aforementioned articles. On the contrary, these studies do not provide data sets comparable to ours. Additionally Sand et al. (2017) and Moschos et. al., (2022) even explicitly point out the lack of data in the central Arctic.

In addition, we have already discussed the articles mentioned by the referee in our manuscript in a critical manner to convey to the reader that while this literature is valuable, there is still a lack of observational data over the central Arctic cryosphere. Here you find the line numbers of our manuscript where we have referred to the aforementioned articles:

a) **Sand et al., 2017** (for modelling)**:**
At line 57-59, "These sparse AOD measurements are used in part to explain AOD variations in climate models (*Sand et al., 2017*; Palazzi et al., 2019). In addition, the lack of AOD measurements in the Arctic, which results in a gap in observational data, further limits our knowledge of aerosol-AA interactions in global and regional climate models (Goosse et al., 2018)."

At line 60-62, "Given the lack of ground-based measurements, several attempts have been made to use AOD, retrieved from the top of the atmosphere reflectance (TOA) observations made by passive satellite remote-sensing instruments over the Arctic, to fill this gap Glantz et al. (2014); Wu et al. (2016); *Sand et al. (2017)*; Xian et al. (2022))."

b) *Schmale et al., 2022* and **Moschos et al., 2022** (for observation)**:**
At line 54-56, "In addition, there are other site-based long-term aerosol measurement studies (Herber et al., 2002; Tomasi et al., 2007; Moschos et al., 2022; Schmale et al., 2022). However, these are not necessarily spatio-temporally representative of the high-latitude Arctic region (Xian et al., 2022)."

In addition, all these three valuable research articles (Sand et al., 2017; Schmale et al., 2022; Moschos et al., 2022) mentioned by the referee have their own scientific focus, which is different from our main focus on the distribution of aerosols over the central Arctic cryosphere, which is rapidly changing due to Arctic warming.

Therefore, the conclusions of our manuscript bring new insights that for the first time provide detailed variability of aerosol load and its composition by integrating high spatiotemporally covered comprehensive observations together with GEOS-Chem model simulations over the central Arctic cryosphere for the first time. Furthermore, we present the seasonal influence of precipitation and smoke plumes intrusion on aerosol distribution and composition over the central Arctic cryosphere .

In order to emphasize the novelty of our work and on our focus we propose to change the following:

**At line 74-77, we propose to modify:**
Furthermore, in addition to ground-based measurements and satellite observations, several valuable studies using models (such as Hardenberg et. al., 2012; Evangeliou et. al., 2016; Sand et. al., 2017; Breider et. al., 2017; Ren et. al, 2020; Schmale et. al., 2021) and reanalysis datasets (such as, Kinne et. al., 2019; Xian et. al., 2021; Xian et. al., 2022) have been conducted over the Arctic. It is worth noting that all of these studies lack observational data, especially over the high Arctic cryosphere region, which is highly vulnerable to warming and climate change in the Arctic (Bintanja et. al., 2017).

**At line 94-96, we propose to modify:**
Consequently, we have extended our objective to examine aerosol composition over the snow- and ice-covered regions of the high Arctic (72-90°N, as defined by Sand et. al.,2017) originating from anthropogenic as well as natural sources with impact of smoke intrusion events and seasonal change in precipitation over the central Arctic cryosphere. To achieve this goal, we investigate aerosol composition in the high Arctic using GEOS-Chem, a global chemical transport model (CTM) coupled with MERRA-2 (Molod et al., 2015) meteorology simulation.

**Q3:** The novelty of this work is therefore not clear to me. In particular, I expect for example that analyses of AOD based on ensembles, as provided by CMIP or AEROCOM would be as relevant as using only one model, at coarse resolution (4°x5°) as done here. The same can be said about reanalyses such as CAMS or MERRA2 which also provide AOD by species/mode at higher spatial and time resolution.
**Response:**
For the novelty of this work we kindly suggest the referee to go through the authors response to the questions Q1(b) and Q2.

Further analysis of CMIP and AeroCom would be an interesting next step, but in our study we used GEOS-Chem 3D Chemical Transport Model coupled with MERRA-2 meteorology for the AEROSNOW retrieval time period, which covers nearly a decade (from the year 2003 to 2011), to study both horizontal and vertical chemical transport to the Arctic with impact of smoke intrusion events and seasonal change in precipitation over the central Arctic cryosphere. The CMIP models do not provide 3D chemical transport of aerosols (Zhao et. al., 2022). Whereas, the AeroCom models simulated only one year (2006) using 2006 meteorology with emissions from the year 2000 (Sand et. al., 2017). GEOS-Chem is coupled with the latest Modern-Era Retrospective Analysis for Research and Applications, Version 2 (MERRA-2) meteorology assimilated with modern hyperspectral radiation and microwave observations and GPS radio occultation datasets. MERRA-2 is well suited for Arctic conditions and minimizes the impact of meteorology on GEOS-Chem model chemistry (Murray et. al., 2021). In addition, we have shown plume intrusion into the Central Arctic cryosphere both horizontally and vertically, as well as aerosol variability with seasonal changes in precipitation over central Arctic sea ice, which is not possible with monthly average outputs from CMIP and AeroCom.

In this study, we performed a global simulation because we consider three-dimensional long-range transport to the Arctic cryosphere, for more than a decade from 2000 to 2012 (13 Years), using three years as the spin-up time. With our computational resources, it took more than three months to complete the simulation, at a horizontal resolution of 4° × 5° (about 440 km × 100 km in the high Arctic latitudes), which meets our needs to study the vast Central Arctic cryosphere. In addition, the average resolution of the CMIP6 models is 2° × 2.5° (Zhao et. al., 2021).

We would like to point out to the referee that we did not use CAMS or MERRA-2 reanalysis data in this study for the following reasons;

    i.  **CAMS:** Copernicus Atmosphere Monitoring Service (Inness et al. (2019) reanalysis dataset of the atmospheric composition produced by the European Centre for Medium-Range Weather Forecasts (ECMWF) assimilates satellite retrievals of AOD from MODIS Terra and Aqua and AATSR (Advanced Along-Track Scanning Radiometer) Envisat, using the ECMWF's Integrated Forecasting System (Inness et al. (2019).

       None of the satellite datasets assimilated by CAMS are retrieved over snow and ice. Therefore, significant errors will be expected. Such a comparison would not be fair.

    ii.  **MERRA-2 reanalysis:** MERRA2 reanalysis which was produced using the Goddard Earth Observing System (GEOS-5; Molod et al. (2015)), for AOD, MERRA2 assimilates data from AVHRR, MODIS, the Multi-angle Imaging SpectroRadiometer (MISR). The algorithms used in all these satellite products are not sensitive for highly bright central Arctic cryosphere (Zhao et. al., 2021).

       The MERRA2 reanalysis data has been assimilated with AVHRR, MODIS, MISR data. We have already discussed in the authors response to the question Q2 that, these valuable space borne datasets has limitations and deficits in optimized algorithms for bright surfaces, which inhibits in providing reliable data over central Arctic snow and sea-ice due to the high reflectivity of these surfaces. So that it does not make much sense for comparison.

**Q4:** Also, the MACv2 aerosol climatology (Kinne 2019) proposes the same information as presented with GEOS-Chem here (monthly AOD by size/species), but based on an ensemble, at 1° resolution, and corrected using AERONET and MAN. Again, I expect that MACv2 does as good a job as the GEOS-Chem simulations presented here. This product (and/or reanalysis/ensembles) should at least be mentioned, and if possible compared to GEOS-Chem in order to make the paper relevant and turn into a multi-model evaluation.

**Response:** We thank the reviewer for this valuable recommendation. We will cite the Max Planck institute Aerosol climatology v1 (MACv2; Kinne et. al., 2019) dataset and would like to take up the suggested comparison in a future publication.

**At line 74-77, we propose to add a citation:**

Furthermore, in addition to ground-based measurements and satellite observations, several valuable studies using models (such as Hardenberg et. al., 2012; Evangeliou et. al., 2016; Sand et. al., 2017; Breider et. al., 2017; Ren et. al, 2020; Schmale et. al., 2021) and reanalysis datasets (such as, Kinne et. al., 2019; Xian et. al., 2021; Xian et. al., 2022) have been conducted over the Arctic. It is worth noting that all of these studies lack observational data, especially over the high Arctic cryosphere region, which is highly vulnerable to warming and climate change in the Arctic (Bintanja et. al., 2017).

**Q5:** In conclusion, although the idea of this work is of interest for the community, I think the novelty and conclusions that can be drawn from it are too limited, and I do not recommend publication in ACP, unless the focus of the paper is changed, with a deeper evaluation and analysis of the information provided by GEOS-Chem to extract novel knowledge on aerosols in sea-ice regions. An idea could be to leverage more the information on the vertical provided by the model.

**Response:**

We feel that in Q1(b), Q2 and Q3 we have presented sufficiently clearly the significance and novelty of our work and ask the *Referee* and *Editor* to consider the arguments carefully.

We regret that the reviewer was mainly focusing on our first objective (satellite-based evaluation) while hardly not considering the second one (scientific significance and application).

I also include specific comments below that could help improve the manuscript.

**Specific comments**

**Q6: Abstract**

L11 - "with a pronounced chemical speciation in GEOS-Chem": what do you mean?

**Response:** "with a pronounced chemical speciation in GEOS-Chem", we mean that GEOS-Chem simulates higher anthropogenic aerosol loading in spring and higher natural aerosol loading in summer over the central Arctic cryosphere.

**At line 11, we propose to modify:**

The space-borne and model simulated AOD show consistent spatiotemporal distributions in both seasons, with a  higher anthropogenic aerosol loading in spring and higher natural aerosol loading in summer in GEOS-Chem simulations.

L20 - seeing as AEROSNOW is built upon AATSR data, which is discontinued since 2012, I do not understand how you would use that for evaluation of aerosol forecast in more recent years. Also, forecast applications are not mentioned elsewhere in the manuscript.

**Response:** Here in this line 20, we mean 'simulation' instead of 'forecast'. We are sorry for the confusion. We will modify in the revised manuscript. Additionally we plan to apply AEROSNOW to SLSTR (Sea and Land Surface Temperature Radiometer, onboard of Sentinel 3a and 3b) which are in orbit since 2016 and 2018. In this respect, we also plan to extend the dataset.

**At line 20, we propose to modify:**

The promising results of AEROSNOW could also serve as the baseline for the evaluation and improvement of aerosol simulations for various chemical transport models, especially over Arctic sea ice.

**Q7: 1 - Introduction**

L36 - Schmale et al., 2021; Pernov et al., 2022

**Response:**  Yes, we agree with the reviewer, we have cited these articles in the revised manuscript.

**At line 36, we propose to add citations:**

The sources of aerosol in the Arctic include long-range transport and changing regional and local emissions of aerosol or its precursors (Schmale et al., 2021; Pernov et al., 2022)

L39 - Li et al., 2022

**Response:**  Yes, we agree with the reviewer, we have cited these articles in the revised manuscript.

**At line 39, we propose to add citations:**

On the other hand, the aerosol that absorbs solar radiation warms the atmosphere and surface (Li et. al., 2022).

L42 - Skiles et al., 2018

**Response:**  Yes, we agree with the reviewer, we have cited these articles in the revised manuscript.

**At line 42, we propose to add citations:**
Carbonaceous aerosols absorb solar radiation when deposited on snow and ice surfaces, reducing the albedo of snow and ice (Skiles et. al., 2018).

L49 - Meinander et al., 2022 (dust);  Lapere et al., 2023 (sea salt); Eck et al., 2009; Marelle et al., 2015 (biomass burning)
**Response:**  Yes, we agree with the reviewer, we have cited these articles in the revised manuscript for dust, sea salt and biomass burning.
**At line 46-49, we propose to add citations:**
During summer (June, July, August, (JJA)), the major source of carbonaceous aerosols in the high Arctic is the result of biomass burning within the Arctic (Eck et al., 2009; Marelle et al., 2015). In contrast, during spring (March, April, and May, (MAM)), long-range transport of emissions, arising from anthropogenic pollution and including biomass burning at mid-latitudes, dominate the sources of aerosol in the Arctic (Marelle et al., 2015). Natural aerosol sources are predominant in the Arctic summer. These sources include wind-blown dust over land, sea salt over the Arctic Ocean, and biomass burning(Meinander et al., 2022; Lapere et al., 2023).

L49 - seeing as you focus on sea-ice covered region in the rest of the paper, you should mention blowing snow sources of sea salt. Even though sea salt does not contribute a lot to AOD, it is still important to mention.
**Response:**  Blowing snow sources of sea salt is prevalent in the Arctic, especially during the winter season when wind strength is high enough (Huang et. al., 2017), but are negligible in the Arctic spring and summer (Huang et. al., 2017).

L51 - "… our understanding of the effect of aerosols on the Arctic climate…"
**Response:**  We have added the term "effect" at line 51 of the revised manuscript.
**At line 51, we propose to modify:**
The Arctic is vast, and the lack of spatio-temporally representative ground-based measurements of AOD limits our understanding of the aerosols effects on the Arctic climate and on AA in particular, and vice versa.

L53 - please use the appropriate reference for the MOSAiC campaign instead of the website address (Shupe et al., 2022 - https://doi.org/10.1525/elementa.2021.00060)
**Response:**  Yes, we agree with the reviewer, we have cited the article in the revised manuscript.
**At line 53, we propose to add citation:**
Recently, several research campaigns/expeditions have taken place. Amongst other objectives selected processes relevant to aerosol formation and loss in the Arctic. Examples are the MOSAiC campaign (Shupe et al., 2022), ACLOUD/PASCAL (Wendisch et al., 2019), PAMARCMIP (Hoffmann et al., 2012; Nakoudi et al., 2018; Ohata et al., 2021).

L57 - this sentence is not clear, what do you mean? please rephrase
**Response:**  The sentence at line 57 of the manuscript  "These sparse AOD measurements are used in part to explain AOD variations in climate models (Sand et al., 2017; Palazzi et al., 2019)", explains that, due to the deficiency of AOD measurements especially over the central Arctic cryosphere the largest uncertainties stem from the direct and indirect effects of aerosols on Arctic clouds and climate, which are poorly represented in models (Sand et al., 2017; Palazzi et al., 2019).
**At line 57, we propose to modify:**
Due to the lack of high spatio-temporal coverage of AOD measurements, especially over the central Arctic cryosphere the largest uncertainties stem from the direct and indirect effects of aerosols on

Arctic clouds and climate, which are poorly represented in models (Sand et al., 2017; Palazzi et al., 2019).

L65 - unclear what you mean, please rephrase

**Response:** We are sorry for the confusion, here at line 65 we mean that the AOD retrieval over Arctic cryosphere is more difficult, because of the large illumination angles. This potentially leads to an overestimation of AOD values (Mei et al., 2013).

**At line 65, We propose to modify:**

In addition, the AOD retrieval over the Arctic cryosphere is more difficult, because of the large illumination angles. This potentially leads to an overestimation of AOD values (Mei et al., 2013).

Please remain neutral in your wording: L71 "successful", L82 "very well explained", L83 "well explained", etc…

**Response:** We propose to remove the words "successful" from line 71, "very well explained" from line 82, and "well explained" from line 83.

**At line 71, we propose to modify:**

In addition, the active sensor, cloud-aerosol lidar with orthogonal polarization(CALIOP/CALIPSO), does not report measurements above 72° N latitude.

**At line 82, we propose to modify:**

AEROSNOW is the algorithm used to retrieve AOD over Arctic snow and ice by using the Advanced Along Track Satellite (AATSR), which is specified in Swain et al. (2023).

**At line 83, we propose to modify:**

The former requires a good knowledge of the retrieval of the AOD distribution over the northernmost Arctic latitudes, which is detailed in (Swain et al., 2023) and a previous study in Istomina et al. (2011).

L83-L84 - "The former …", "The latter …" - unclear what this refers to, please rephrase.

**Response:** At the line 83-84, we mean "the former" as space borne retrieval and "the latter" as GC model simulations.

**At line 83-84, we propose to modify:**

The space-borne retrieval of aerosol loading over the central Arctic cryosphere requires a good understanding of the development of an advanced retrieval algorithm, which is described in detail in (Swain et al., 2023) and an earlier study in Istomina et al. (2009, 2011). In addition, a chemical transport model is required for the breakdown of the total aerosol load determined by satellite into different aerosol components with up-to-date emission inventories.

L84-85 - what is an "optimal selection of AOD components"?

**Response:** We suggest to modify the sentence.

**At line 84-85, we propose to modify:**

In addition, a chemical transport model is required for the breakdown of the total aerosol load determined by satellite into different aerosol components with up-to-date emission inventories.

L86 - please include a reference for the mid-latitude product you say is available.

**Response:** We would like to cite Li et. al., (2019) in line 86.

**At line 86, we propose to add citation:**

Such a component-based aerosol loading using passive satellite remote sensing measurements is currently available only for mid-latitudes and not for the Arctic (Li et al., 2019).

L87-93 - repletion of previous points - I suggest moving this part before L50 and merging with paragraph L45

**Response:** We believe that the paragraph at line number 87-93 is at right place, because of the second objective of our manuscript, which is the important explanation before introducing the second objective of this manuscript that is to examine aerosol composition over the snow- and ice-covered regions of the high Arctic, which is driving the total aerosol load over this sub-region with impact of seasonal precipitation and smoke intrusion.

The paragraph at line 87-93 of our manuscript is presented as follows:
"As mentioned earlier, long-range transport of emissions from forest fires (Sand et al., 2017; McCarty et al., 2021) is an important source of carbonaceous aerosol, i.e., BC and OC in the Arctic. Changes in these components will alter the AOD and lead to changes in the radiative forcing (Stone et al., 2014).

According to recent studies (Sherstyukov and Sherstyukov, 2014; Hugelius et al., 2020; McCarty et al., 2021), biomass burning in the low Arctic will increase in the future. In this regard, peat thaw in Siberia is also of potential importance. Efforts to better understand the total AOD and corresponding aerosol components, especially in the vulnerable high Arctic, are therefore timely."

L103 - the AEROSNOW AOD data was evidently already generated in Swain et al. (2023a), so this sentence does not belong in this paper

**Response:** Yes, the data was already generated in Swain et al. (2023), but we need to cite the paper in this manuscript as well because we are using the datasets here.

**Q8: 2 - Data sets and data processing**

L114 - why/how? please explain

**Response:** In the line 114 "Comparison of AOD in these two seasons enables the impact of the long-range transport and local aerosol sources to be identified and investigated (Willis et al., 2018)".

This line explains that as per Willis et. al., 2018, it is clear that Arctic experience aerosol load from long-range transport during spring, whereas natural local origin is prominent during summer due to the receding of Arctic dome during summer. Hence, by investigating these two seasons will identify the impact of them on aerosol load over central Arctic cryosphere.

**At line 114, we propose to modify:** To investigate the distribution, variability, and remote and local sources of Arctic aerosols over snow and ice, we have used passive remote sensing in spring (March-April-May, MAM) and summer (June-July-August, JJA). Comparison of AOD in these two seasons enables the impact of the long-range transport and local aerosol sources to be identified and investigated (Willis et al., 2018).

Figure 1 - the same figure appears in Swain et al., 2023a - please mention it in the caption.

**Response:** We propose to cite Swain et al., 2023 at Figure 1. Location of PEARL(80.054°N, 86.417°W), OPAL(79.990°N, 85.939°W), Hornsund(77.001°N, 15.540°E) and Thule(76.516°N, 68.769°W) AERONET measurement stations considered in this study (Swain et al., 2023).

**We propose to add citation to title of Figure 1:**
Figure 1. Location of PEARL(80.054°N, 86.417°W), OPAL(79.990°N, 85.939°W), Hornsund (77.001°N, 15.540°E) and Thule(76.516°N, 68.769°W) AERONET measurement stations considered in this study (Swain et al., 2023).

L152-154 - this is not relevant to explain here, please remove.
**Response:** Yes, we agree with the reviewer.
**At line 152-154, we propose to remove:**
The formation of sulfate aerosol occurs by the oxidation of SO2. Initially OH reacts with SO2 to make HSO3, which reacts with O2 to make an HO2 and SO3. The latter reacts in the gas phase with H2O to form H2SO4.

L155-158 - I suggest moving this text into a table instead. This is a bit wordy.
**Response:** We would like to keep the line 155-158 as it is, as we have moved entire emission inventories used "section 2.2.1" to Appendix B of this manuscript.

Section 2.2.1 - this section has too much detail. I suggest moving most of it to an Appendix or supplementary material, and keep only keep information on the emission inventories directly into section 2.2
**Response:** Yes, we agree with referee.
We propose to move Section 2.2.1 the information about emission inventories used in our study to Appendix B of the revised manuscript.

**Q9: 3 - Results: Evaluation of AODs from AERONET, AEROSNOW, and GEOS-Chem**

Figure 2 - AEROSNOW and AERONET data are already presented in the exact same way in Figure 5 of Swain et al., 2023a. Please mention it in the caption.
**Response:** Yes, we agree with the referee.
**We propose to add citation to the caption of Figure 2:**
Figure 2. Monthly mean time-series of GC AOD, speciated and natural, and the AERONET and AEROSNOW measured AOD (Swain et. al., 2023) at PEARL,OPAL, Hornsund and Thule stations. The MAM,JJA periods are highlighted with blue shades. Annotations for each time series show rmse and R between AEROSNOW and AERONET AODs.

L218 - R is not defined
**Response:** We have defined R in the revised manuscript at line 218 as follows "Here AEROSNOW and GC agree with AERONET datasets well, with Pearson correlation coefficient (R) values being 0.90."
**At line 218, we propose to modify:**
Here, AEROSNOW and GC agree well with the AERONET measurements, with the Pearson correlation coefficient (R) values of 0.90 and 0.90, with confidence intervals (CI) of 0.57, 0.98 and 0.42, 0.84, and associated p-values of 1.5e-4 and 5.9e-6, respectively.

L218/219 - I would be more nuanced: the correlation coefficients are only computed over a handful of points and for monthly averages. Therefore it is not surprising to obtain this kind of correlation. Also, a confidence interval/level of significance of these correlations would be appreciated given the limited number of data points.
**Response:**
The significance of the correlation is tested using the Monte Carlo method. The associated p-value is used to determine whether or not the correlation is significant within a 95% confidence interval (CI). The p-value roughly indicates the probability that an uncorrelated system will produce data sets that have a Pearson correlation at least as extreme as that calculated from those data sets. We have added the confidence interval and significance level of these correlations in the revised manuscript in the Appendix A figures, such as Figure A1. A2, A3 and A4.

**At line 2018/219, we propose to modify:**
In general, the time series for AERONET is well reproduced by both AEROSNOW and GC. The best agreement is for the PEARL AERONET station. Here, AEROSNOW and GC agree well with the AERONET measurements, with the Pearson correlation coefficient (R) values of 0.90 and 0.90, with confidence intervals (CI) of 0.57, 0.98 and 0.42, 0.84, and associated p-values of 1.5e-4 and 5.9e-6, respectively. The associated p-value is used to determine whether or not the correlation is significant within a 95% confidence interval. The p-value roughly indicates the probability that an uncorrelated system will produce data sets that have a Pearson correlation at least as extreme as that calculated from these data sets.

L219 – "comparatively"
**Response:**  We are sorry for the typo.
**At line 2019, we propose to correct the typo:**
The latter is surprising bearing in mind, the comparatively low spatial resolution of GC model results as compared to the point measurement made at AERONET stations.

L227 "a detailed description is given in Part-I of this work". Is Part-I Swain et al., 2023a?
**Response:**  Yes, we are sorry for the confusion.
**At line 227, we propose to modify:**
Having gained confidence in the satellite dataset (a detailed description is given in Swain et. al., 2023), we now investigate the GC model results.

L233 - R is defined here for the first time but already used in L218
**Response:**  We have defined R in line 218 of the revised manuscript and would like to remove from line 233.
**At line 218, we propose to modify:**
Here AEROSNOW and GC agree with AERONET datasets well, with Pearson correlation coefficient (R) values being 0.90.

Figure A1 - this exact figure is already in Swain et al., 2023a as Figure A1.
**Response:**  Yes, we agree with the referee.
**We propose to add citation to title of Appendix Figure A1 :**
Figure A1. Validation of monthly mean AEROSNOW retrieved AOD colocated with monthly mean AERONET observation AOD obtained over PEARL, OPAL, Hornsund, and Thule stations. The linear regression lines are shown as blue dashed lines (Swain et al., 2023).

L240 - this is a bit of a stretch in my opinion. The speciation given by GEOS-Chem is not validated in this paper. As a result, you could get the right total AOD for the wrong reasons via error compensation across species. It is not possible to conclude that you have the right emissions/optical properties, only the right AOD. I would be more careful and nuanced about this type of conclusions.
**Response:** The aerosol speciation of GEOS-Chem is well validated by Breider et al. (2017). Further, line 240 " We conclude that accurate emission inventories and optical properties were used in GC, adequately representing the true AOD over these observation sites." is already in result section of our study and we found comparatively good spatio-temporally seasonal agreement between GC model, satellite retrieved and ground based measurements, which is shown in Fig.3 of this manuscript. We would like to modify the line 240 as follows, "We conclude that recent updated emission inventories and optical properties were used in GC, adequately representing the true AOD over these observation sites.

**At line 240, we propose to modify:**
We conclude that the up-to-date emission inventories and optical properties were used in GC are adequately representing the true AOD over these observation sites.

L241 - same comment, plenty of other reasons could explain the discrepancies, other than spatial resolution (biases in meteorology e.g.). Besides, it is unclear how you do the comparisons: do you compare only for times/days where you have data in all datasets, or do you just take all the available data for all. This can also strongly affect the results since there are probably many missing data points in observations (night-time, non-clear sky…) whereas GC provides "continuous" data. Ideally you should compare only over times with data available in all data sets, otherwise you get a sampling bias.
**Response:** Yes we agree with the referee, the biases in GC AOD can be attributed to many reasons, including how well the model presents new particle formation. Yes, in this study we have compared all datasets colocated with time.
**At line 241, we propose to modify:**
The remaining biases of GC AOD can be attributed also to many other reasons, including deficiencies in new particle formation and effects of coarse horizontal model resolution (4°x5°).

Figure 4 - the color code is misleading with same colors for fine/coarse and BC/Sulfate. Please correct.
**Response:** Yes, we have corrected the color code of Fig.4 and Fig.8.
The fine and coarse mode particle colors of Fig.4 and Fig.8 has changed from red, blue to purple and yellow respectively in the revised manuscript.

L252 - I do not understand what 11% and 7% refer to. Please clarify this.
**Response:** The line 252 "The CM AOD shows variation with 11% and 7% between the sites, located over CA and over Hornsund during MAM and JJA respectively." shows the variation of coarse mode particles between AERONET sites located in Canadian Archipelago (CA) and Hornsund AERONET site located on Spitsbergen are 11% and 7% during spring (MAM) and summer (JJA) respectively.
**At line 252, we propose to modify:**
The variation of coarse mode particles between AERONET sites located in Canadian Archipelago (CA) and Hornsund AERONET site located in Spitsbergen are 11% and 7% during spring and summer respectively.

L255-256 - or to problems of emissions/optical properties/meteorology - cf my previous comment on that.
**Response:** At the line 255-256 "The climatological difference in the FM and CM AOD between AERONET and GC might be traced back to a potential overestimation of haze events of GC during spring."

Yes, the meteorology certainly has an impact on chemistry in model, but in our study we have used GC simulations coupled with most recent MERRA-2 meteorology assimilated with modern hyperspectral radiance and microwave observations, along with GPS-Radio Occultation datasets, which is very suitable for Arctic conditions as it is assimilated with observational data, but still we cannot completely rule out the impact of meteorology on aerosol load.

L265 - "we suspect" - could you do dedicated simulations with/without forest fire emissions to prove that?

**Response:** At the line 265 "We suspect that the increase in OC and the larger presence of FM AOD at PEARL, OPAL, and Thule indicate the impact by boreal forest fires. This finding agrees with that from (Sand et al., 2017; Xian et al., 2022)". As explained in the author's response to question number Q3, in this study, we have used global simulation to show the 3D transport to Arctic cryosphere, as we are considering three dimensional long-range transport in this study, for more than one decade from 2000 to 2012 (13 years) by keeping three years as spin-up time and it took more than three months for the global simulation to complete with our computational resources. For this reason we want to avoid bulky runs of the model.

L266-267 - I agree this is the most likely cause, but there is no evidence here. Please rephrase.
**Response:** We agree with the reviewer.
**At line 266-267, we propose to modify:**
This finding agrees with that from Sand et al., (2017) and Xian et al., (2022). The boreal forest fires impact on carbonaceous aerosol (BC+OC) load is also seen in the contour maps in appendix Fig.A5.

Figure A5 - I am used to seeing AOD as an integrated value over the vertical, i.e. a 2D, not 3D field. What does a vertical profile of AOD mean? This should be explained in the data section to help the reader.
**Response:** We are sorry for the typo mistake, it is the the vertical Extinction Coefficient of carbonaceous aerosols per layer.
**We propose to modify the title of Figure A5:**
Vertical zonal mean of Extinction Coefficient of carbonaceous aerosols (BC+OC) per layer for (a) MAM and (b) JJA respectively, averaged from the year 2003 to 2011. The black box shows the biomass burning within the Arctic.

**Q10: 4 - Arctic AOD climatology and boreal forest fires**

L296 - are these secondary aerosols really important for AOD? What about sea salt re-emissions from blowing snow processes? Could that account for the missing AOD?
**Response:**
Yes, the formation of new particles in the warming Arctic is very important specifically over the central Arctic sea-ice, which is experiencing rapid change and creating a large open ocean emissions, as most of the models are unable to capture it (Boucher et. al., 2013; Creamean et. al., 2022).

Blowing snow sources of sea salt is prevalent in the Arctic, especially during the winter season when wind strength is high enough (Huang et. al., 2017), but are negligible in the Arctic spring and summer (Huang et. al., 2017). Further, sea-salt is very much hygroscopic in nature and easily gets wet scavenged in rain dominated central Arctic sea-ice region (Bintanja et. al., 2017).

L319 - based on what? the emission inventory in GC? General observations from the literature?
**Response:** The line 317-319 "The increase of BC+OC during summer, when long-range transport from the mid-latitudes is not significant, confirms the relevance and penetration of Arctic boreal forest fires into the high Arctic sea ice covered areas (Fig. 10). The black box in Fig. A5 shows the latitudinal range in which forest fire originate."

This is based on the GC simulations and presented in Fig.10 and Appendix Fig.A5 of the manuscript.

**Q11: 5 - Conclusions**

In general the conclusions do not bring any new knowledge about aerosols in the Arctic, they only confirm that GC behaves reasonably, in accordance with what is already known.
**Response:**
We kindly request the referee to consider the authors response to Q1(b), Q2, and Q3 for better insights of the conclusions of this manuscript.

    **i.** First ever confirmation of observational aerosol distribution over central Arctic cryosphere for the entire cycle of sea-ice growth and decline, which shows that AEROSNOW retrieved space-borne observations can serve as a baseline for not only the GEOS-Chem model but also for various other models over this region of the Arctic.

    **ii.** The conclusion of the manuscript presents a large-scale analysis focusing on the spatiotemporal coverage with regional, seasonal, and annual variations of the total aerosol load and its components from anthropogenic and natural sources. In addition, the effects of smoke intrusion events that change aerosol composition from more sulfate-dominated aerosol to carbonaceous aerosols are examined. Seasonal changes in precipitation over the Central Arctic cryosphere influence the seasonal aerosol load over the Central Arctic cryosphere. This confirms mechanisms such as precipitation and smoke intrusion into the central Arctic cryosphere influence the total aerosol distribution determined by AEROSNOW for the cycle of sea ice growth and decline, as well as the annual variations for the period of almost a decade (from 2003 to 2011).

L332-335 - again, I think it is very important to note that points (i) and (ii) mentioned in this sentence are already presented in Swain et al., 2023a and are not new to the present work. Only (iii) is a novelty.
**Response:**
Points (i) and (ii) (as well as (iii)) address the aforementioned two objectives. Points (i)-(iii) shall be understood as building blocks of the integrated study which is reported in this manuscript.
We have hopefully clarified our point of view in our response to Q1(b), Q2, and Q3.

This study has two objectives, which are explained in the author's response to question Q1(b). The first objective achieved (evaluation of the model) is listed below in point (i) and the second objective achieved in point (ii), while in item (iii) some suggestions for future improvements in the simulation of aerosol loading over the central Arctic were discussed in our manuscript.

    **i.** **AOD retrieval over Arctic snow and ice:** In this point of the conclusion we discussed the time-series of AOD retrieved by AEROSNOW over the central Arctic cryosphere surface for almost a decade (2003-2011) time span and used to evaluate global atmospheric 3-D chemical transport using GC. Further, we discussed about the high anthropogenic aerosol loading (Arctic Haze events) due to long-range transport over Arctic snow and ice is captured by the AOD behavior in the AEROSNOW AOD as well as that of GC. The time series and seasonality of the GC, AEROSNOW, and AERONET AOD agree well.

    **ii.** **Arctic AOD climatology:** In this point of the conclusion, we thoroughly discussed about the spatio-temporal distributions and variations from AEROSNOW and GC, with seasonal distribution of the contribution of anthropogenic aerosols to total AOD is dominant in spring, while naturally occurring aerosols predominate in summer. Further, we also have discussed the GC simulated impact of seasonal precipitation and biomass burning smoke is

changing the total composition of aerosol from more sulfate dominated aerosol to carbonaceous aerosols over this rapidly changing sub-region of the Arctic.

iii. **Overall performance of the AEROSNOW retrieval and GEOS-Chem model simulations:** In this point of the conclusion, we finally discussed the overall performance of the AEROSNOW and GEOS-Chem, and discussed some suggestions for the further improvements such as, this study provides a potentially new dataset of AOD over the Arctic. We conclude that improved meteorology and emission inventories for central Arctic sea ice during spring and summer will improve the accuracy of the GC AOD. This will in turn facilitate the assessment of the AA and the related sea ice loss.

Thus, these points in the conclusion of this manuscript are completely different from the Swain et al., 2023, brings interesting scientific insites with higher significance for the scientific community for aerosol variability over rapidly changing central Arctic cryospheric region.

L336-345 - several sentences are very similar to the conclusions in Swain et al., 2023a, sometimes word for word.
**Response:** The paragraph between lines 336-345 of this ACP manuscript is the comparison between AEROSNOW, GEOS-Chem, and AERONET datasets, which is why it sounds similar but not the same as lines 256-257 of our AMT paper.

Such as the line 340-342 in our ACP manuscript *"The high anthropogenic aerosol loading (Arctic Haze events) due to long-range transport over Arctic snow and ice is captured by the AOD behavior in the AEROSNOW AOD as well as that of GC. The time series and seasonality of the GC, AEROSNOW, and AERONET AOD agree well."* is bit similar with line number 256-257 of our AMT paper *"The high anthropogenic aerosol loading (Arctic haze events) due to long-range transport over Arctic snow and ice is captured by the AOD determined by AEROSNOW. The time series and seasonality of the AEROSNOW AOD agree well with AERONET observations."*

Here we propose to modify the line 340-342 in our ACP manuscript to remove the similarity between ACP and AMT papers.
**At line 340-342, we propose to modify:**
Arctic Haze events, which occur during the spring season due to long-range transport of anthropogenic aerosols to the Arctic (Willis et al., 2018), are also captured by the AOD determined by GC and AEROSNOW. The good agreement was also observed when comparing the time series and seasonality of GC, AEROSNOW, and AERONET AOD.

L361 - "(cite some articles here)". Please review more carefully your manuscript before submission.
**Response:** We are very sorry for this leftover from the late phase of manuscript preparation.
In this bracket we wanted to write: "Willis et al., 2018; Xian et al., 2022".
**At line 361, we propose to add citation:**
The combination of hydrophobicity and increasing boreal forest fires means that carbonaceous aerosols (black carbon, BC, and OC) are an increasingly important contributor to total AOD over Arctic sea ice in summer (Willis et. al., 2018, Xian et. al., 2022).

L370 - I believe this is contradictory with your assessment that emissions are correct earlier (L240)
**Response:** At line 370, we actually mean that by using improved emission inventories with respect to receding sea ice which is exposing more open ocean emission will further improve the accuracy of AOD simulations by the models.

**At line 370, we propose to modify:**
We conclude that by using improved input meteorology and natural oceanic emissions due to rapidly declining sea-ice exposing more open ocean for central Arctic region during spring and summer could improve the accuracy of the GC AOD simulations.

**References:**

- Swain, Basudev, Marco Vountas, Adrien Deroubaix, Luca Lelli, Yanick Ziegler, Soheila Jafariserajehlou, Sachin S. Gunthe, and John P. Burrows. "Spring and summertime aerosol optical depth retrieval over the Arctic cryosphere by using satellite observations." Atmospheric Measurement Techniques Discussions 2023 (2023): 1-19. https://amt.copernicus.org/preprints/amt-2023-65/.

- Von Hardenberg, J., L. Vozella, C. Tomasi, V. Vitale, A. Lupi, M. Mazzola, T. P. C. Van Noije, A. Strunk, and A. Provenzale. "Aerosol optical depth over the Arctic: a comparison of ECHAM-HAM and TM5 with ground-based, satellite and reanalysis data." Atmospheric Chemistry and Physics 12, no. 15 (2012): 6953-6967.https://doi.org/10.5194/acp-12-6953-2012.

- Boucher, Olivier, David Randall, Paulo Artaxo, Christopher Bretherton, Gragam Feingold, Piers Forster, V-M. Kerminen et al. "Clouds and aerosols." In Climate change 2013: The physical science basis. Contribution of working group I to the fifth assessment report of the intergovernmental panel on climate change, pp. 571-657. Cambridge University Press, 2013.

- Schmale, Julia, Paul Zieger, and Annica ML Ekman. "Aerosols in current and future Arctic climate." Nature Climate Change 11, no. 2 (2021): 95-105. https://www.nature.com/articles/s41558-020-00969-5.

- Evangeliou, Nikolaos, Yves Balkanski, Wei Min Hao, Alexander Petkov, Robin P. Silverstein, Rachel Corley, Bryce L. Nordgren et al. "Wildfires in northern Eurasia affect the budget of black carbon in the Arctic–a 12-year retrospective synopsis (2002–2013)." Atmospheric Chemistry and Physics 16, no. 12 (2016): 7587-7604. https://doi.org/10.5194/acp-16-7587-2016.

- Sand, Maria, Bjørn H. Samset, Yves Balkanski, Susanne Bauer, Nicolas Bellouin, Terje K. Berntsen, Huisheng Bian et al. "Aerosols at the poles: an AeroCom Phase II multi-model evaluation." Atmospheric Chemistry and Physics 17, no. 19 (2017): 12197-12218. https://doi.org/10.5194/acp-17-12197-2017.

- Breider, Thomas J., Loretta J. Mickley, Daniel J. Jacob, Cui Ge, Jun Wang, Melissa Payer Sulprizio, Betty Croft et al. "Multidecadal trends in aerosol radiative forcing over the Arctic: Contribution of changes in anthropogenic aerosol to Arctic warming since 1980." Journal of Geophysical Research: Atmospheres 122, no. 6 (2017): 3573-3594. https://doi.org/10.1002/2016JD025321.

- Ren, Lili, Yang Yang, Hailong Wang, Rudong Zhang, Pinya Wang, and Hong Liao. "Source attribution of Arctic black carbon and sulfate aerosols and associated Arctic surface warming

during 1980–2018." Atmospheric Chemistry and Physics 20, no. 14 (2020): 9067-9085. https://doi.org/10.5194/acp-20-9067-2020.

- Schmale, Julia, Paul Zieger, and Annica ML Ekman. "Aerosols in current and future Arctic climate." Nature Climate Change 11, no. 2 (2021): 95-105. https://www.nature.com/articles/s41558-020-00969-5.

- Xian, Peng, Jianglong Zhang, Norm T. O'Neill, Travis D. Toth, Blake Sorenson, Peter R. Colarco, Zak Kipling et al. "Arctic spring and summertime aerosol optical depth baseline from long-term observations and model reanalyses–Part 1: Climatology and trend." Atmospheric Chemistry and Physics 22, no. 15 (2022): 9915-9947. https://doi.org/10.5194/acp-22-9915-2022.

- Xian, Peng, Jianglong Zhang, Norm T. O'Neill, Jeffrey S. Reid, Travis D. Toth, Blake Sorenson, Edward J. Hyer, James R. Campbell, and Keyvan Ranjbar. "Arctic spring and summertime aerosol optical depth baseline from long-term observations and model reanalyses–Part 2: Statistics of extreme AOD events, and implications for the impact of regional biomass burning processes." Atmospheric Chemistry and Physics 22, no. 15 (2022): 9949-9967. https://doi.org/10.5194/acp-22-9949-2022.

- Tomasi, Claudio, Alexander A. Kokhanovsky, Angelo Lupi, Christoph Ritter, Alexander Smirnov, Norman T. O'Neill, Robert S. Stone et al. "Aerosol remote sensing in polar regions." Earth-Science Reviews 140 (2015): 108-157. https://doi.org/10.1016/j.earscirev.2014.11.001.

- Schmale, Julia, Sangeeta Sharma, Stefano Decesari, Jakob Pernov, Andreas Massling, Hans-Christen Hansson, Knut Von Salzen et al. "Pan-Arctic seasonal cycles and long-term trends of aerosol properties from 10 observatories." Atmospheric Chemistry and Physics 22, no. 5 (2022): 3067-3096. https://doi.org/10.5194/acp-22-3067-2022.

- Creamean, Jessie M., Kevin Barry, Thomas CJ Hill, Carson Hume, Paul J. DeMott, Matthew D. Shupe, Sandro Dahlke et al. "Annual cycle observations of aerosols capable of ice formation in central Arctic clouds." Nature communications 13, no. 1 (2022): 3537. https://www.nature.com/articles/s41467-022-31182-x.

- Bintanja, Richard, and Olivier Andry. "Towards a rain-dominated Arctic." Nature Climate Change 7, no. 4 (2017): 263-267. https://www.nature.com/articles/nclimate3240.

- Moschos, Vaios, Julia Schmale, Wenche Aas, Silvia Becagli, Giulia Calzolai, Konstantinos Eleftheriadis, Claire E. Moffett et al. "Elucidating the present-day chemical composition, seasonality and source regions of climate-relevant aerosols across the Arctic land surface." Environmental Research Letters 17, no. 3 (2022): 034032. DOI 10.1088/1748-9326/ac444b.

- Zhao, Alcide, Claire L. Ryder, and Laura J. Wilcox. "How well do the CMIP6 models simulate dust aerosols?." Atmospheric Chemistry and Physics 22, no. 3 (2022): 2095-2119. https://doi.org/10.5194/acp-22-2095-2022.

- Murray, Lee T., Eric M. Leibensperger, Clara Orbe, Loretta J. Mickley, and Melissa Sulprizio. "GCAP 2.0: a global 3-D chemical-transport model framework for past, present, and future climate scenarios." Geoscientific Model Development 14, no. 9 (2021): 5789-5823. https://gmd.copernicus.org/articles/14/5789/2021/

- Inness, Antje, Melanie Ades, Anna Agustí-Panareda, Jérôme Barré, Anna Benedictow, Anne-Marlene Blechschmidt, Juan Jose Dominguez et al. "The CAMS reanalysis of atmospheric composition." Atmospheric Chemistry and Physics 19, no. 6 (2019): 3515-3556. https://doi.org/10.5194/acp-19-3515-2019

- Molod, Andrea, Lawrence Takacs, Max Suarez, and Julio Bacmeister. "Development of the GEOS-5 atmospheric general circulation model: Evolution from MERRA to MERRA2." Geoscientific Model Development 8, no. 5 (2015): 1339-1356. https://gmd.copernicus.org/articles/8/1339/2015/

---

## Author Comment (AC2)

**Referee #2**

**Previous title:** Spring and summertime aerosol optical depth variability over Arctic cryosphere from space-borne observations and model simulation

**Revised title:** Variability of aerosol over the Arctic cryosphere from space-borne observations and model simulations for the cycle of sea ice growth and decline.

The authors thank the referee for her/his effort, and time taken to review our manuscript. The valuable criticisms and comments have helped us to improve our paper. We hope that we have been able to answer satisfactorily the questions raised and clarify parts of the manuscript which were unclear or ambiguous.

We have changed the title of the manuscript following the recommendation of referee.

In the following the referee comments and criticisms, our responses, as authors, and our resultant changes to the manuscript are colored **black**, **blue** and **red** respectively.

In this study, the authors have used the total aerosol optical depth (AOD) determined by the AEROSNOW algorithm and data from the AATSR satellite instrument over snow- and ice-covered regions of the Arctic. The dataset is used to evaluate the global GEOS-Chem 3D chemical transport model for the period 2003-2011.

The retrievals over bright surfaces are associated with large uncertainties because the main contribution to the signal comes from the surface and not from atmosphere, which is optically thin in Arctic in majority cases. Therefore, I appreciate the work performed by the authors in the evaluation and intercomparison of the retrieved AOT with the global GEOS-Chem 3D chemical transport model for the extended period of time.

I advice that the authors improve the paper. The paper can be reconsidered after major revision. Some comments are given below:

**Q1:** What is the definition of Q_ext in Eq.(2)? Is it /? is the average extinction cross section of particles, is the average projected area of the particles.
**Response:** The $Q_{ext}$ is defined as extinction efficiency ($Q_{ext}$). The extinction efficiency are calculated using refractive index and lognormal size distribution data available from the Global Aerosol Data Set (GADS) (Köpke et. al., 1997; Chin et. al., 2002).
**At line 176, we propose to add the definition of ($Q_{ext}$):**
Where, $Q_{ext}$ is defined as the extinction efficiency. The extinction efficiency is calculated using refractive index and lognormal size distribution data available from the Global Aerosol Data Set (GADS) (Köpke et. al., 1997; Chin et. al., 2002). The column mass loading and the particle mass density are presented as M and ρ respectively (Tegen and Lacis, 1996).

**Q2:** I would advice to change the title of this paper. The title is similar to the title of the paper under review located at https://amt.copernicus.org/preprints/amt-2023-65/amt-2023-65.pdf. I do not think that it is a good idea to have several identical figures in both papers. The identical figures must be removed from this paper.

**Response:** Yes, we agree with the reviewer, we have changed the title from "Spring and summertime aerosol optical depth variability over Arctic cryosphere from space-borne observations and model simulation" to "Variability of aerosol over the Arctic cryosphere from space-borne observations and model simulations for the cycle of sea ice growth and decline".

We would like to cite the two figures we used in this ACP manuscript from our other paper that is under review in AMT: "Swain et al., 2023a. Spring and summertime aerosol optical depth retrieval over the Arctic cryosphere by using satellite observations (doi:10.5194/amt-2023-65).

These two figures are captioned:

  i) Figure.1, with title Location of PEARL(80.054°N, 86.417°W), OPAL(79.990°N, 85.939°W), Hornsund (77.001°N, 15.540°E) and Thule(76.516°N, 68.769°W) AERONET measurement stations considered in this study".

  ii) Appendix Figure A1. With title "Validation of monthly mean AEROSNOW retrieved AOD colocated with monthly mean AERONET observation AOD obtained over PEARL, OPAL, Hornsund, and Thule stations. The linear regression lines are shown as blue dashed line".

**We propose to modify the title of the manuscript:**
Variability of aerosol over the Arctic cryosphere from space-borne observations and model simulations

**We propose to add a citation to title of Figure 1 and Appendix Figure A1 :**
Figure 1. Location of PEARL(80.054°N, 86.417°W), OPAL(79.990°N, 85.939°W), Hornsund(77.001°N, 15.540°E) and Thule(76.516°N, 68.769°W) AERONET measurement stations considered in this study (Swain et al., 2023).

Figure A1. Validation of monthly mean AEROSNOW retrieved AOD colocated with monthly mean AERONET observation AOD obtained over PEARL, OPAL, Hornsund, and Thule stations. The linear regression lines are shown as blue dashed lines (Swain et al., 2023).

**References:**

- Chin, Mian, Paul Ginoux, Stefan Kinne, Omar Torres, Brent N. Holben, Bryan N. Duncan, Randall V. Martin, Jennifer A. Logan, Akiko Higurashi, and Teruyuki Nakajima. "Tropospheric aerosol optical thickness from the GOCART model and comparisons with satellite and Sun photometer measurements." Journal of the atmospheric sciences 59, no. 3 (2002): 461-483. https://doi.org/10.1175/1520-0469(2002)059.

- Koepke P., M. Hess, I. Schult, and E.P. Shettle, "Global aerosol dataset", Report N 243, Max-Plank-Institut für Meteorologie, Hamburg, 44 pp., September 1997. https://hdl.handle.net/21.11116/0000-0009-EB9B-0